# Spontaneous persistent activity and inactivity in vivo reveals differential cortico-entorhinal functional connectivity

Krishna Choudhary[1,9], Sven Berberich[2,3], Thomas T. G. Hahn[4], James M. McFarland[5] & Mayank R. Mehta ● [1,6,7,8] ✉

Understanding the functional connectivity between brain regions and its emergent dynamics is a central challenge. Here we present a theory-experiment hybrid approach involving iteration between a minimal computational model and in vivo electrophysiological measurements. Our model not only predicted spontaneous persistent activity (SPA) during Up-Down-State oscillations, but also inactivity (SPI), which has never been reported. These were confirmed in vivo in the membrane potential of neurons, especially from layer 3 of the medial and lateral entorhinal cortices. The data was then used to constrain two free parameters, yielding a unique, experimentally determined model for each neuron. Analytic and computational analysis of the model generated a dozen quantitative predictions about network dynamics, which were all confirmed in vivo to high accuracy. Our technique predicted functional connectivity; e. g. the recurrent excitation is stronger in the medial than lateral entorhinal cortex. This too was confirmed with connectomics data. This technique uncovers how differential cortico-entorhinal dialogue generates SPA and SPI, which could form an energetically efficient working-memory substrate and influence the consolidation of memories during sleep. More broadly, our procedure can reveal the functional connectivity of large networks and a theory of their emergent dynamics.

Cognition requires the interaction between several large neural networks, each network containing millions of neurons, each neuron in turn characterized by many microscopic parameters. To study the complex emergent properties of systems with large degrees of freedom, the statistical physics approach is to develop a quantitative model, based on only the salient order parameters, and subsequently test its predictions quantitatively, not just qualitatively, in simplified experimental preparations that capture the essence of the emergent principles. In line with this tradition, we develop an analytically tractable model of spontaneous activity in interacting neural networks, and quantitatively verify several predictions of the theory in vivo during default, internally generated activity in the absence of external sensory stimuli.

During quiescence, deep sleep, under anesthesia, and in vitro, local neural networks from many brain areas, including cortex, show nearly synchronous, rhythmic activity termed delta oscillations, non-

[1]Department of Physics and Astronomy, University of California, Los Angeles, Los Angeles, CA, USA. [2]Department of Psychiatry and Psychotherapy, Central Institute of Mental Health, Medical Faculty Mannheim, Heidelberg University, Mannheim, Germany. [3]Department of Psychiatry and Psychotherapy, University Medical Center, Johannes Gutenberg University, Mainz, Germany. [4]Zentralinstitut fur Seelische Gesundheit, Mannheim, Germany. [5]Department of Physics, Brown University, Providence, RI, USA. [6]W. M. Keck Center for Neurophysics, University of California, Los Angeles, Los Angeles, CA, USA. [7]Department of Electrical and Computer Engineering, University of California, Los Angeles, CA, USA. [8]Departments of Neurology and Neurobiology, University of California, Los Angeles, Los Angeles, CA, USA. [9]Present address: HRL Laboratories, Malibu, CA, USA. ✉e-mail: mayankmehta@ucla.edu

REM sleep oscillations, slow wave sleep (SWS) etc[1–5]. The membrane potential ($V_m$) of individual neurons exhibits nearly synchronous transitions between the depolarized and more active (Up), and less active (Down) states[1,5,6]. This is reflected in the local field potential (LFP), which too shows Up-Down state oscillations (UDS) synchronous with the membrane potential[6–9]. The UDS are ubiquitously found across several species and experimental preparations and are considered the default activity of many networks[10–13]. Several studies have suggested that the interactions between cortical regions during UDS are crucial for stimulus response[7,14], memory consolidation[15–19]. Impairment of UDS causes learning and memory deficits, while UDS enhancement leads to improvement[14,20,21].

While most cortical areas show synchronous UDS oscillations[17], recent in vivo studies have shown that pyramidal neurons in layer 3 of the medial (MECIII), but not lateral (LECIII), entorhinal cortex show spontaneous persistent activity (SPA) during UDS: events where the neuron's $V_m$ persists in the depolarized Up state while the afferent neocortical areas transition to the Down state[22]. This definition differs from other studies that define singular Up states, within an isolated network, as a form of persistent activity[23]. Instead, it is reminiscent of the persistent activity during working memory tasks, which shows sustained activation after the stimulus is extinguished. Network models show that such sustained activity during awake, working memory tasks can be generated through reverberant excitation and feedback inhibition, but it is unclear whether these models can explain spontaneously evoked persistent activity[24,25]. Depolarizing current injections elicit sustained activity in vitro[26–28] but do not elicit SPA within MECIII neurons during UDS, implicating network rather than intracellular mechanisms[22]. Currently, network models of UDS employ an attractor framework with two fixed points, one active (the Up state) and one inactive (the Down state), with adaptation driving the oscillation[29–33]. Such models, however, have not been used to understand large interacting networks, and are thus agnostic about major experimental findings, like the quantization of SPA during UDS[34]. Furthermore, existing theories focus exclusively on the active state, discarding the inactive state as simply a recovery phase for network adaptation. However, the energy function of discrete Hopfield networks[35,36] is symmetric under activity inversion ($+1 \rightarrow -1$), so the physics suggests that, just like the active states, these inactive states are themselves energy minima in the landscape and could thus also be utilized as a memory substrate.

Here we show that a simple, mean-field model involving two interacting networks of excitation-inhibition can capture the observed dynamics of SPA during UDS. Our theory also exploited the symmetric inactive attractor to predict, to our knowledge, a new phenomenon: spontaneous persistent inactivity (SPI). To test the model quantitatively, we used the in vivo cortico-entorhinal circuit as our model system. Anatomically, the neocortex serves as an afferent source of input to several subcortical regions, including the entorhinal cortex[37,38]. To measure neocortical ensemble activity during UDS, we recorded the extracellular LFP from the parietal cortex. As the parietal cortex receives strong inputs from most neocortical regions[39–42] and UDS is synchronous across all neocortical areas[2,7,17,43], this LFP acted as the afferent reference for neocortical UDS. Simultaneously, we did whole-cell $V_m$ measurements from anatomically identified pyramidal neurons in various efferent nearby cortical areas, including frontal (FRO) and prefrontal (PRE) cortices, and distant efferent regions in the entorhinal cortices (EC). To establish a baseline for comparison, we also recorded from several neurons in the parietal (PAR) cortex, close to the LFP recording site. As the spontaneous activity of single neurons is tightly linked to the cortical networks in which they are embedded, this allowed us to probe the activity of localized networks within each target region[44]. Within the EC, the medial (MEC) and lateral (LEC) subdivisions are anatomically and functionally distinct: the MEC contains spatially selective "grid cells"[45], while the LEC is thought to

encode objects or experienced time[46–51]. We focused on the EC layer 3 (MECIII/LECIII) regions, since MECIII neurons are a major source of input to the hippocampus, show SPA in vivo, and are crucial in the generation and maintenance of UDS in the MEC[22,52].

We found significant in vivo SPA in MECIII and SPI in both MECIII and LECIII, but not in the neocortical regions FRO and PRE with respect to PAR. We developed an iterative procedure to quantitatively match computational simulations of our minimal model with experimental observables, which in turn led to a general theory of network-network interactions that made several falsifiable predictions. All predictions, from the relative timing of state transitions to the quantized, history-dependent nature of SPA and SPI, were confirmed with in vivo data. Further, our results attributed the differences in SPA and SPI across cells to differences in feed-forward excitatory connectivity from the large neocortical network to the specific efferent subnetwork and the recurrent excitatory connectivity within the efferent subnetwork itself. This prediction too was verified using publicly available connectomics data. The number of experimental observations explained by our theory are far greater than the number of model parameters we varied, demonstrating its predictive power and generality. To our knowledge, our study is the first to predict theoretically and detect experimentally the novel phenomena of persistent inactivity, and show that both SPA and SPI not only co-occur but are the result of common network interaction principles and can provide an estimate of the functional connectivity between large networks.

## Results

### The mean field model of cortical interaction predicts both spontaneous persistent activity (SPA) and inactivity (SPI)

A minimal mean field network supporting UDS has three biologically well-established ingredients: excitatory neurons, inhibitory neurons, and activity-dependent adaptation of the excitatory (but not inhibitory) neurons[29–31,33]. We constructed a mean field model of two cortical regions, each with its own recurrently connected inhibitory and excitatory populations[53,54] (Fig. 1). In isolation, each network exhibits transitions between Up and Down states (Supplementary Figs. 1, 2) that are the stable fixed points of the dynamical system of equations, much like local minima in an energy landscape[55,56]. Their stability is inversely related to their distance from the separatrix, a line which defines the boundary between the basins of attraction. The slowly-varying, activity-dependent adaptation translates the excitatory nullcline, thus influencing the stability of each state. Growing adaptation governs the transition from the Up to Down state, while external drive and a falling adaptation governs the transition from Down to Up. Underlying gaussian noise gives the network a non-zero "temperature," preventing it from stagnating in a particular state for arbitrarily long time periods. For simplicity, quantitative falsifiability, and based on available observations, we assumed that all internal parameters except the recurrent excitation strength $W_{INT}$ are identical across the afferent and efferent networks.

Importantly, these two networks are connected only unidirectionally, with the afferent network sending an excitatory projection $W_{EXT}$ to the excitatory population in the efferent network, with no connection backwards from efferent to afferent (Fig. 1a). The UDS oscillation of the afferent network rhythmically destabilizes the endogenous UDS oscillation of the efferent network (Fig. 1b, c, Supplementary Fig. 3). Larger values of $W_{EXT}$ lead to phase locking (Fig. 1d), i.e. the efferent network merely follows the afferent network. Smaller values of $W_{EXT}$, on the other hand, give rise to situations where the efferent network does not follow the afferent UDS, resulting in transient desynchronizations between the two networks[57]. Under weak drive $W_{EXT}$ from the afferent network, an increase in the efferent recurrent excitatory connectivity ($W_{INT}$) drives the efferent Up state fixed point away from the separatrix, increasing the Up state stability. This was verified using simulations, which demonstrate a novel,

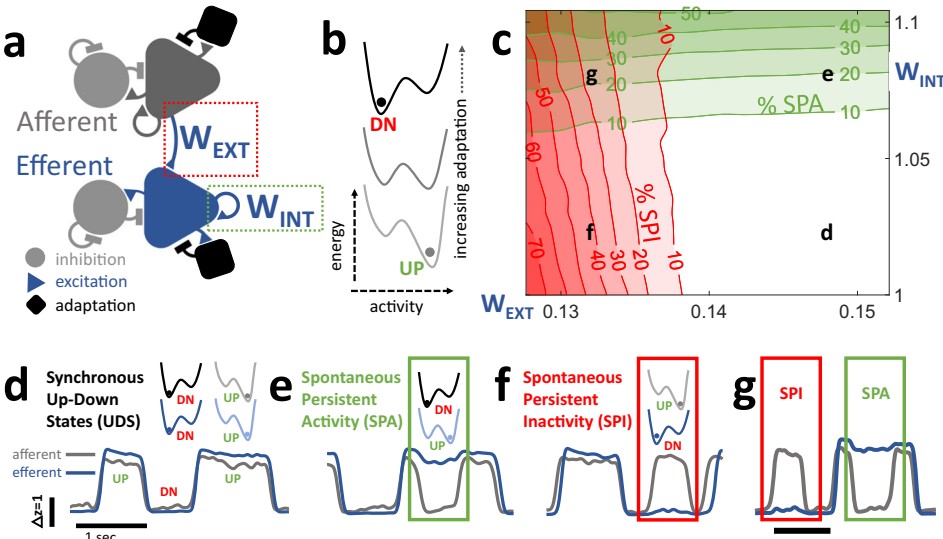

**Fig. 1 | A simple mean-field model predicts *spontaneous persistent activity and inactivity* in the efferent network. a** The model consists of two networks, each characterized by the average activity of excitatory (*triangle*) and inhibitory (*circle*) populations, with only excitation showing activity dependent adaptation (*square*). The afferent network (*gray*) provides excitatory input (red box: $W_{EXT}$) to the efferent (*blue*) network, but not vice versa. All internal parameters are identical between the two networks, except the recurrent excitation (green box: $W_{INT}$). **b** Each network is described by a potential energy landscape with two local minima, corresponding to the Up and Down states. The activity-dependent changes in adaptation level influences the stability of each minima, creating the Up-Down state (UDS) oscillation. **c** By modulating $W_{INT}$ and $W_{EXT}$, the model introduces transient desynchronizations between the two networks. There are four distinct regimes ((**d**–**g**) in the four corners of the plot): **d** With high $W_{INT}$ and low $W_{EXT}$, the model can reproduce synchronized UDS in the two networks (afferent in gray, efferent in

blue), with neither SPA nor SPI. The same scale bars for amplitude (in z-score) and time (1s) are used for (**d**–**f**). **e** Increasing efferent $W_{INT}$ increases the stability of the efferent Up state attractor, and leads to *spontaneous persistent activity* (*SPA*, green box), when the efferent network gets "stuck" or persists in the Up state even when the afferent makes a transition to the Down state. Green contours in (**c**) show time-averaged SPA rates in simulations of the parameter space. **f** Conversely, decreasing $W_{EXT}$ decreases the size of destabilizing afferent current and leads to *spontaneous persistent inactivity* (*SPI*, red box), when the efferent network persists in the Down state while the afferent makes a transition to Up state. Red contours in (**c**) show SPI rates. **g** With lower $W_{EXT}$ and higher $W_{INT}$, the same network can exhibit *SPA* and *SPI* at different times. The positions of each example in the 2D parameter plane are shown in (**c**). Throughout the figures in this text, the green and red colors will denote data/parameters related to SPA and SPI, respectively.

network mechanism to generate *SPA*: instances when the efferent network remains in the Up state, skipping one or more afferent Down states (Fig. 1e). These could explain the experimentally reported *SPA*. Further, a decrease in the strength of $W_{EXT}$ decreases the destabilizing effect of the afferent transitions on the efferent network; the efferent then remains in a Down state, skipping one or more afferent Up states. We call this phenomenon *spontaneous persistent inactivity* (*SPI*: Fig. 1f). The model further predicts that *SPA* and *SPI* are relatively independent, as increasing $W_{INT}$ while simultaneously decreasing $W_{EXT}$ gives rise to coupled UDS sequences exhibiting both *SPA* and *SPI* (Fig. 1g). Notably, the model predicts that $W_{EXT}$ is nearly ten times smaller than $W_{INT}$. Furthermore, a less than 10% change in $W_{EXT}$ ($W_{INT}$) causes a much larger, all or none change in the SPI (SPA) probabilities.

### Detection of *SPA* and *SPI* in MECIII and LECIII in vivo

To monitor network interactions during spontaneous activity in vivo, mice were lightly anesthetized with urethane to induce robust and steady UDS that were synchronous across the entire neocortex. A hidden Markov model was used to classify the data into a binary UDS sequence[58]. Consistent with previous studies, the neocortical LFP and the $V_m$ of neurons in the neocortical regions PAR ($N = 24$), FRO ($N = 7$), and PRE (N-14), and the efferent regions MECIII ($N = 50$) and LECIII ($N = 16$) showed clear bimodal UDS (Fig. 2, Supplementary Fig. 4). For subsequent analysis, the rate of *SPA* (*SPI*) was defined as the proportion of efferent Up (Down) states which outlasted an entire afferent Down (Up) state during an entire experiment. As a first test of the model, we computed the relationship between the neocortical LFP and the $V_m$ from PAR pyramidal neurons, which were recorded close by (0.5 mm apart), and other neocortical neurons in FRO and PRE. These regions are highly connected to

other neocortical regions[7,17,39–43], so $W_{EXT}$ is large, and the model predicts complete phase locking (Fig. 1c), with virtually nonexistent *SPA* and *SPI*. This was indeed the case (Figs. 2b, 3a, Supplementary Fig. 4).

Consistent with previous studies, MECIII neurons showed clear instances of *SPA* (Figs. 2c, 3a), while LECIII neurons did not[22]. In contrast, both LECIII and MECIII neurons showed clear instances of the newly predicted *SPI* (Figs. 2d, 3a). Our model also predicted relative independence of *SPA* and *SPI*; consistently, some MECIII neurons showed both *SPA* and *SPI*, only a few seconds apart (Fig. 2e), and levels of *SPA* and *SPI* within the population of LECIII and MECIII neurons were not significantly correlated (Supplementary Fig. 5). Finally, *SPA* and *SPI* levels were not correlated with the duty cycle and the frequency of neocortical UDS, indicating that they were not artifacts of differences in brain states across experiments (Supplementary Fig. 6).

### Testing the hypothesis that variation in $W_{EXT}$ and $W_{INT}$ alone can explain most in vivo differences between and within MECIII and LECIII

The properties of *SPA* and *SPI* not only varied across brain regions, but even between different neurons from the same region (Fig. 3a). Following our dynamical systems analysis, we hypothesized that all of these differences could arise from just two network parameters: the strength of recurrent excitation in the efferent network ($W_{INT}$) and the strength of external excitatory input to the efferent network ($W_{EXT}$). To test this idea, we used a two-step approach. First, we simulated all possible networks in this 2D parameter space by varying only $W_{EXT}$ and $W_{INT}$, while leaving all other variables unchanged. Modulating just two free parameters yielded networks with a wide range of both *SPA* and *SPI*. Thus, we could estimate the two crucial network variables, $W_{INT}$

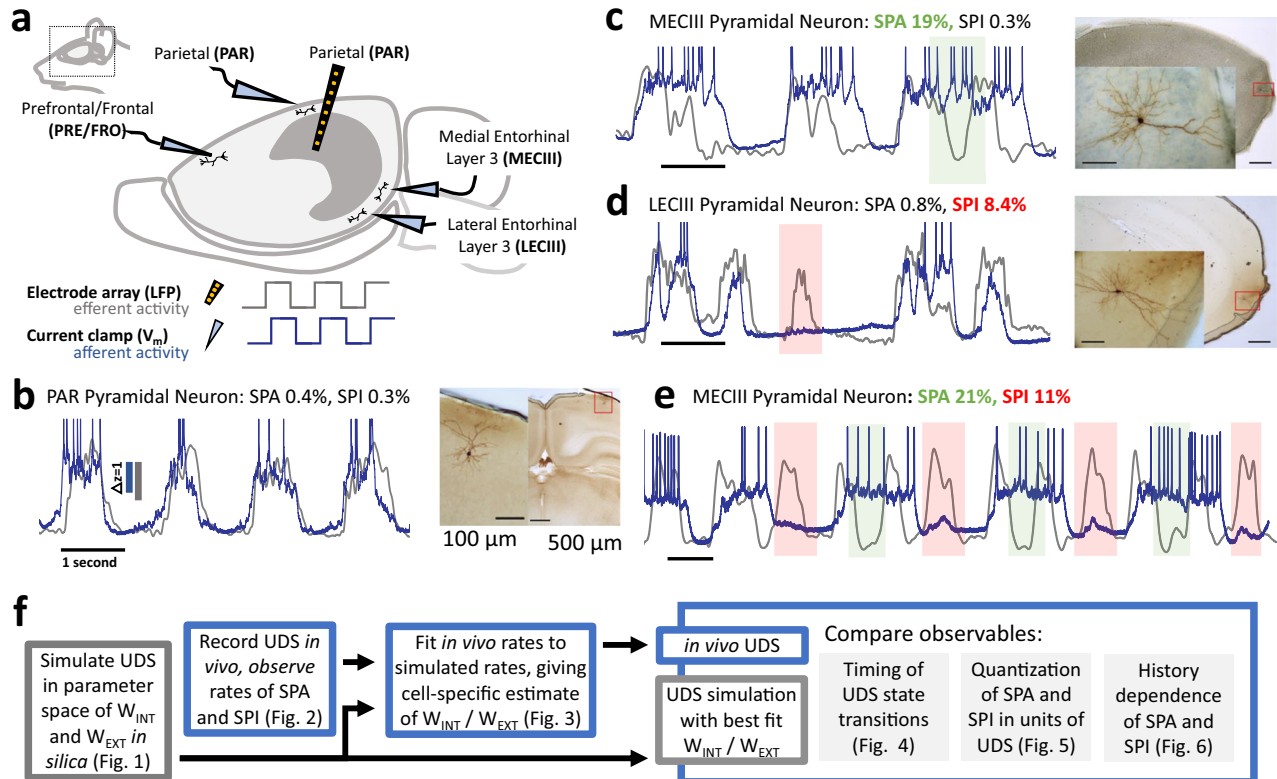

**Fig. 2 | In vivo, simultaneous whole-cell patch clamp and multi-electrode array confirms prediction of SPA and SPI in the entorhinal cortex. a** Experimental design: mice were anesthetized to induce UDS, and local field potential (LFP) from the parietal cortex was measured using a silicon probe (black). Simultaneously, membrane potential ($V_m$) was measured using whole-cell patch clamp from an anatomically identified neuron. The parietal LFP was treated as the afferent reference representing neocortical activity (gray in **b**–**e**), and the $V_m$ traces constituted efferent activity (blue in **b**–**e**). **b** PAR neuron's $V_m$ was phase-locked to the neocortical LFP, matching theory (Fig. 1d). Action potentials have been truncated for clarity. The same scale bars for amplitude (z-scores) were used throughout, and time scale bars (1s) are shown for each individual experiment. The identified UDS sequence is shown above the traces, with histological reconstructions (right) of brain region and the patched cell (insets). Scale bars in histological figures correspond to 100 μm for higher magnification (insets), 500 μm for lower magnification. **c** Clear *SPA* (green box) in the $V_m$ of an MECIII pyramidal neuron, matching (Fig. 1e). **d** Clear *SPI* (red box) in the $V_m$ of an LECIII pyramidal neuron, matching (Fig. 1f). **e** Both *SPA* and *SPI* exhibited by the same MECIII pyramidal neuron, like (Fig. 1g). **f** Schematic of our technique to obtain a quantitative test of the theory using the experimental data.

and $W_{EXT}$, by simply computing the amount of *SPA* and *SPI* observed experimentally and matching those numbers with the appropriate simulation (Fig. 3b, Supplementary Fig. 7). Crucially, even though SPA and SPI varied by 1000-fold across different neurons, in both simulations and in experimental data, this large variation in *SPA* and *SPI* rates could be closely captured by varying only two physiological variables in the model. Clearly, the experimental data were impacted by dozens of additional variables too, e.g. the strength of recurrent inhibition, inputs from other brain regions, synaptic and ionic dynamics etc., that are not included in the model, so this strong fit was unexpected. The robustness of this method was confirmed by using an alternate fitting procedure, which yielded very similar fits between the simulations and in vivo data for each neuron (Supplementary Fig. 8).

The model postulated that *SPA* probability should grow exponentially with increasing internal recurrence $W_{INT}$, and that *SPI* probability should diminish exponentially with increasing external input $W_{EXT}$ (Fig. 3c). Not only were both of these predictions confirmed across all brain areas, but the exponential relationship also spanned nearly 3 orders of magnitude. Further, according to the model, dependence of *SPA* on $W_{INT}$ (SPI on $W_{EXT}$) should be stronger than on $W_{EXT}$ (SPI on $W_{INT}$), but still significant and similarly exponential. This prediction too was confirmed by the data (Fig. 3c). The tightness of the exponential relationship, even though other parameters were not considered, suggests that these two connectivity parameters are indeed the key driving factors governing *SPA* and *SPI* in vivo.

While *SPA* and *SPI* prevalence across neurons was uncorrelated, perhaps due to many other extraneous factors (e.g. depth of anesthesia), the fitted values of $W_{INT}$ and $W_{EXT}$, which seem insensitive to these variables, were significantly negatively correlated, indicating differential properties of the networks (Supplementary Fig. 5). Intuitive trends between parameters were preserved: e.g. neurons with greater net excitatory input ($W_{EXT} + W_{INT}$) should have higher firing rate, and this was confirmed in vivo, as data showed greater mean firing rates for MECIII than LECIII, even at the level of individual cells (Supplementary Fig. 9). This is unexpected since the neurons in the model are mean-field, not spiking. Further, LECIII neurons' $V_m$ (Up: −67.4 ± 2.3 mV, Down: −79 ± 1.9 mV) was significantly less depolarized than MECIII neurons ($V_m$ (Up: −52.4 ± 1.1 mV, Down: −74.2 ± 0.98 mV), further confirming the model predictions.

The two-parameter model, thus constrained by experiments, made major predictions about the nature of large-scale connections between and within these brain regions. Briefly, our model implies that neocortical input into ECIII is weaker than into other neocortical regions, like parietal, frontal, and prefrontal cortices (Fig. 3f). Further, it predicts that recurrent excitation within MECIII is significantly stronger than within LECIII (Fig. 3e). These predictions too are supported by other experiments in vivo and in vitro (see Discussion). Additionally, several further predictions of the model could be tested using the in vivo data from neocortical-LFP/neuron-$V_m$ recordings and the matched simulation of an afferent/efferent connected network system.

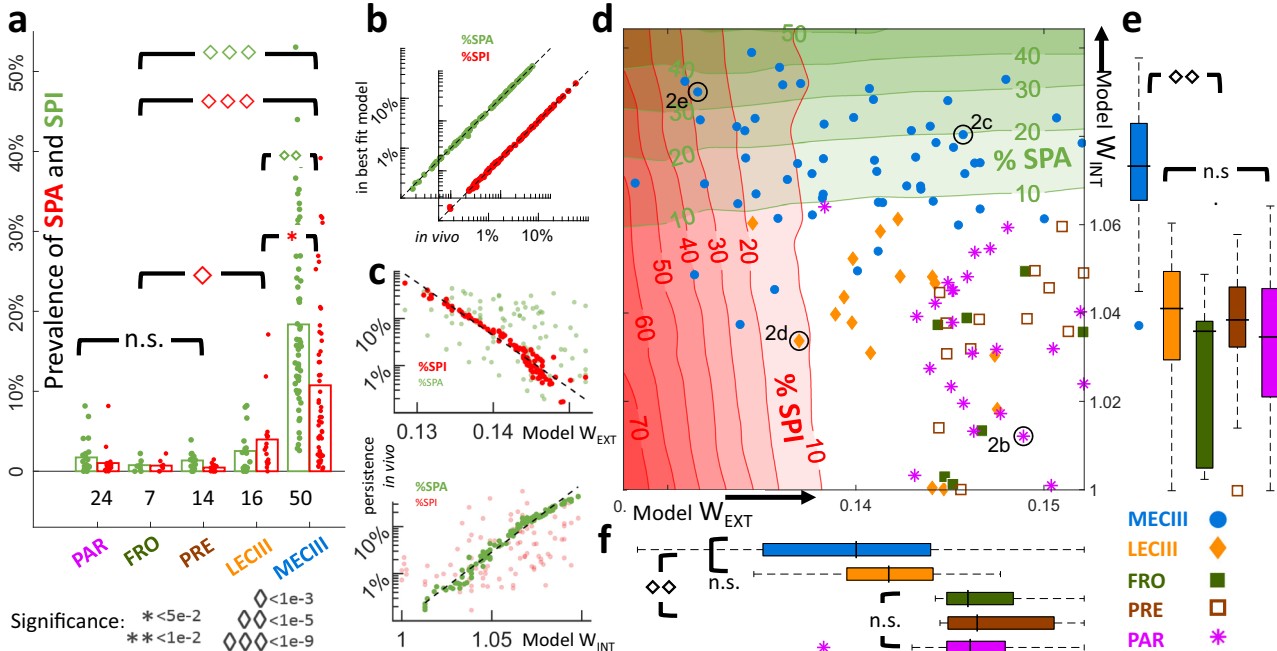

**Fig. 3 | Just two parameters, $W_{EXT}$ and $W_{INT}$, can quantitatively explain neuron-neuron and region-region differences to reveal differential cortico-MECIII vs cortico-LECIII connectivity. a** SPA and SPI rates varied significantly across brain regions, with highest SPA (mean ± SEM: 18.5 ± 7.5%) and SPI (10.2 ± 5.3%) in MECIII, diminished SPA (3.3 ± 2.4%) and SPI (5.9 ± 3.1%) in LECIII. PAR (SPA: 2.6 ± 1.8%, SPI: 1.9 ± 0.7%), FRO (SPA: 0.9 ± 0.3%, SPI: 0.9 ± 0.3%), and PRE (SPA: 1.9 ± 0.4%, SPI: 0.6 ± 0.3%) showed even smaller rates. MECIII SPA and SPI rates were both significantly greater than LECIII rates (SPA: $p < 10^{-5}$, SPI: $p = 0.03$) and neocortical rates (SPA: $p < 10^{-9}$, SPI: $p < 10^{-9}$), while for LECIII only the SPI rate was significantly larger from neocortex ($p < 10^{-5}$). No significant differences across neocortical areas. The number of neurons recorded within each brain region is indicated, and individual neurons are scattered above each bar. **b** SPA and SPI rates were used to match experiments and simulations. **c** Top: Increasing $W_{EXT}$ exponentially decreased the rate of SPI in the model (SPI∝e^{-Wext/0.004}, r = −0.96). Similarly for SPA probability, but the effect is weaker (r = −0.52). Bottom: Increasing $W_{INT}$ exponentially increased the

probability of SPA in the model (SPA∝e^{Wint/0.015}, $r = 0.98$). Similarly for SPI, but weaker ($r = 0.54$). **d** Results of the fitting procedure, after which each cell was assigned a connectivity estimate ($W_{EXT}$ and $W_{INT}$). The background contours and shaded regions are identical to those in Fig. 1c, showing the time-averaged rates of SPA% (green contours) and SPI% (red contours) in simulations. Each cell is represented by a point, and the example cells used in Fig. 2b–e are circled in black. **e** The fitting predicted that $W_{INT}$ was the largest for MECIII (1.073 ± 0.01), while LECIII (1.04 ± 0.01) and neocortical regions were significantly ($p < 10^{-9}$) lower. **f** The fitting revealed that $W_{EXT}$ from the neocortex was the largest for other neocortical areas (0.14621 ± 0.0051), smaller for LECIII (0.14215 ± 0.012), and smallest for MECIII (0.14 ± 0.023). $W_{EXT}$ to entorhinal areas was significantly smaller than to other neocortical areas ($p < 10^{-5}$). Box edges indicate the 25th and 75th percentile, and center black bar denotes the median. Significant differences were established using two-sided nonparametric Wilcoxon rank-sum tests for equal medians.

## Inferred cortico-entorhinal connectivity predicts differential, state-dependent latency to UDS transitions in MECIII vs LECIII

The latency between the up-down sequences in the afferent and efferent networks presents another opportunity to test the model's validity. There are two additional observables: the Up-Down transition delay and the Down-Up transition delay. These latencies differ across brain regions (Fig. 4a, Supplementary Fig. 10). Since neurons behave like leaky capacitors, the strength of afferent excitatory input should be inversely correlated with the response latency of the efferent neurons[59,60]. Therefore, the model predicts that the neurons with larger strength of afferent input $W_{EXT}$ should respond sooner to the neocortical Down-Up transitions, i.e. smaller latency between neocortical LFP and the neuron's $V_m$ (Fig. 4b–d). Consistent with this prediction, LECIII cells with greater predicted excitatory input $W_{EXT}$ showed significantly shorter Down-Up transition latency. A similar result was found within the population of MECIII neurons. Further, consistent with model prediction that $W_{EXT}$ from the neocortex to LECIII is stronger than to MECIII, the population of LECIII neurons showed shorter Down-Up latency than the MECIII population. While $W_{EXT}$ enhances the coupling between the two networks, larger values of $W_{INT}$ make the efferent network more independent of the input. The effect of these competing inputs is state dependent, differentially modulating the efferent Down-Up vs. Up-Down transitions. During an afferent Down-Up transition, the efferent network is in the Down state, where recurrent excitation $W_{INT}$ does not contribute. Thus, the latency

of the efferent Down-Up transition should be relatively insensitive to $W_{INT}$ but depend strongly on $W_{EXT}$. This prediction too was supported across both LECIII and MECIII cell populations (Supplementary Fig. 11).

According to the theory, the situation is reversed for the Up-Down transition: when the efferent network is in the Up state, the recurrent excitation $W_{INT}$ contributes strongly and helps sustain the Up state despite the loss of afferent input, which is in the Down state. Networks with higher $W_{INT}$ have more stable Up states, thereby increasing their "inertia." Thus, the model predicts that ECIII neurons with greater predicted $W_{INT}$ should follow the neocortical Up-Down transitions with longer latency. This was confirmed for both MECIII and LECIII across cells (Fig. 4f, g). In contrast to Down-Up transitions, the latency of the efferent Up-Down transition should be relatively insensitive to $W_{EXT}$ compared to $W_{INT}$. This prediction too was supported across individual neurons within MECIII, within LECIII, and across the MECIII vs LECIII ensemble. The model does predict a small but significant effect, as higher $W_{EXT}$ enhances coupling and decreases latency. This prediction too was supported by our analysis (Supplementary Fig. 11).

These latencies were more correlated with the predicted $W_{INT}$ and $W_{EXT}$ values than with simply the levels of SPA or SPI (Supplementary Fig. 11), further supporting the model. The model also makes quantitative predictions for Down-Up and Up-Down latencies for each cell, and these are highly correlated with the experimental value obtained for each cell (Fig. 4e, h). While the latencies in our model can differ from the experimental observations by a constant value due to

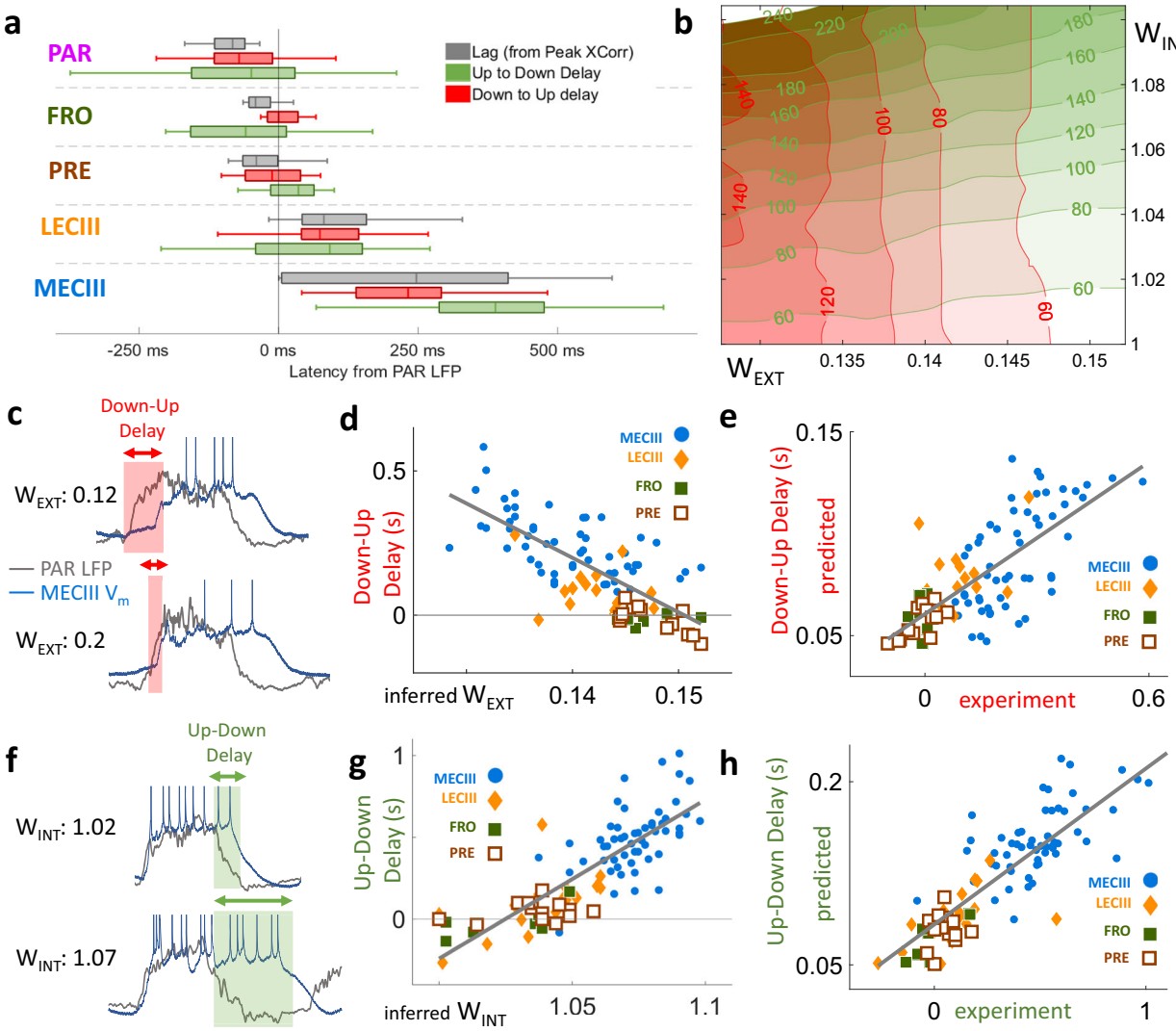

**Fig. 4 | Model predicts transition delays in the efferent network based on differences in recurrent ($W_{INT}$) and external ($W_{EXT}$) inputs. a** The latency between the afferent UDS (parietal *LFP*) and efferent UDS ($V_m$) was quantified using 3 measures: cross-correlation (gray), delay between up-to-down state transitions (green), and delay between down-to-up transitions (red). Parietal $V_m$ preceded the nearby LFP, while other neocortical areas (FRO and PRE) were concurrent. Entorhinal areas (LECIII and MECIII) were significantly delayed from PAR LFP. The number of data points in each region can be found in Fig. 3a. Box edges indicate the 25th and 75th percentile, and center black bar denotes the median. The numerical values can be found in the Supplementary Table 1. **b** Similar to Fig. 1c, but for state transition delays in simulations of the model parameter space. The theory predicts that modulating $W_{INT}$ and $W_{EXT}$ has differential effects on the Down-Up delay (red contours) and Up-Down delay (green contours). Increasing $W_{EXT}$ increases the influence of afferent transitions on the efferent network, thus decreasing the Down-Up delay between networks. Increasing the internal excitation $W_{INT}$ of the efferent network stabilizes the efferent Up attractor, prolonging persistence in the Up state, thus decreasing the Up-Down delay. **c** Examples from two experiments show the neuron with larger $W_{EXT}$ had smaller Down-Up delay (red shaded area). **d** Each neuron's average Down-Up transition delay in vivo was significantly anti-correlated with the predicted value of $W_{EXT}$ given by the fitted model ($r = -0.66$, $p < 10^{-15}$). **e** The in vivo Down-Up transition delay was significantly correlated with the predicted delay from the matched simulation ($r = 0.726$, $p < 10^{-26}$). **f** Examples from two experiments show the neuron with larger predicted value of $W_{INT}$ having larger Up-Down transition delay. **g** Each neuron's average Up-Down transition delay in vivo was significantly correlated with the predicted value of $W_{INT}$ ($r = 0.823$, $p < 10^{-30}$). **h** The in vivo Up-Down transition delay was significantly correlated with the predicted latency from the matched simulation ($r = 0.847$, $p < 10^{-16}$). Correlations and $p$-values were computed using Spearman's rank correlation coefficient.

processes we did not model, e.g., the finite velocity of signal propagation in brain tissue, the difference in latency across neurons should be well matched. Further, the propagation delay could be different for Up states vs. Down states. We estimated this delay for each neuron using the difference between the experimentally observed transition delay and the predicted delay from our model (Supplementary Fig. 12). We found that the Up state and Down state propagation delays were highly correlated on a cell-by-cell basis, and that MECIII Down state delays were longer than Up state delays. This hints at a yet-unknown mechanism that keeps MECIII neurons in the Up-state for even longer than the higher $W_{INT}$ value can explain. The predicted model parameters $W_{EXT}$ and $W_{INT}$ were more strongly correlated with the UDS

latencies than with the mean firing rates (Supplementary Fig. 9), further supporting the model and ruling out nonspecific effects.

### *SPA* and *SPI* are quantized by neocortical UDS both in vivo and in the model, reflecting an underlying history-dependent Bernoulli process

The model predicts that *SPA* and *SPI* are all-or-none events that are initiated and terminated by state transitions in the afferent network. As a result, even though efferent Up(Down) state and *SPA*(*SPI*) durations form a continuous, unimodal distribution, these durations should be quantized in integral units of the afferent UDS cycles (Fig. 5a, Supplementary Fig. 13). To visualize this for *SPA*, segments of the

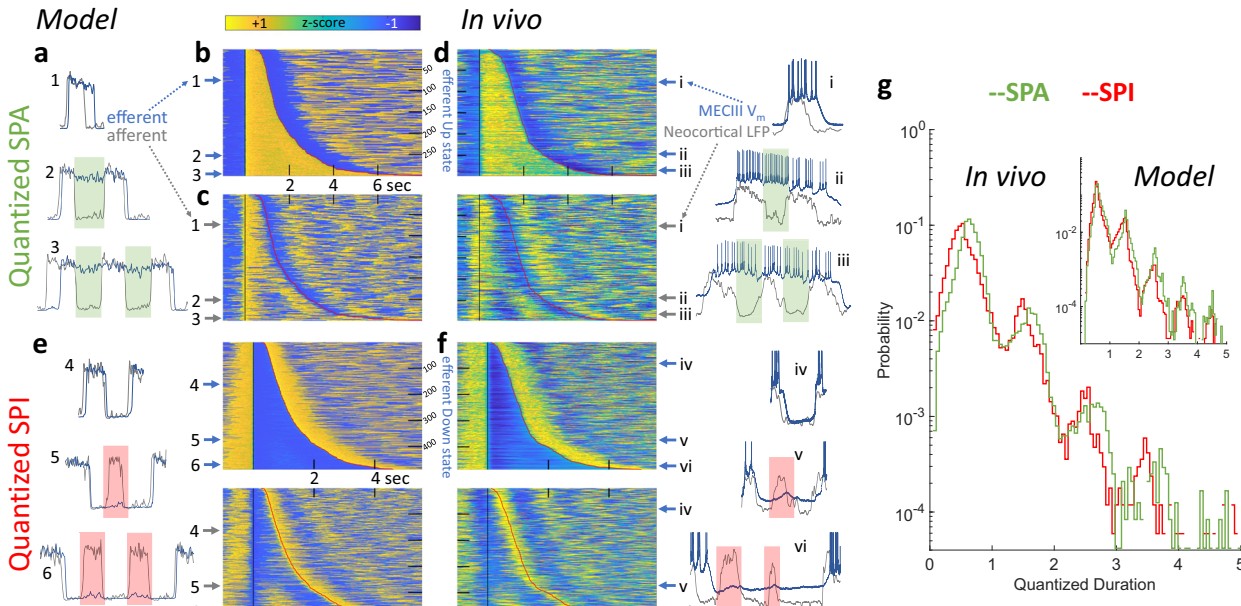

**Fig. 5 | *SPA* and *SPI* are quantized in units of afferent neocortical UDS cycles. a** In the model, the efferent *SPA* (blue) span a continuous range of durations, but they are quantized (green boxes, 1, 2, 3) in the units of afferent UDS cycles (gray). **b** Using simulated UDS, all efferent Up states were aligned to the efferent Down-Up transition and ordered with increasing duration. Each row represents a single efferent Down-Up transition. **c** A second matrix was constructed using the same time points, but using the afferent network activity. The examples (1-3) in (**a**) correspond to the row numbers in (b/c). Efferent *SPA* can last 8 s, spanning several afferent UDS cycles. Same color axis shows amplitude (in z-score) for all the panels. **d** In vivo data from an MECIII neuron, matched to the simulation used in (**a**–**c**), validates the model by showing efferent MECIII Up states (blue) that last integer multiples (i-iii)

of afferent neocortical LFP UDS cycles (gray). **e** Similar to (**a**–**d**), but for *SPI*, showing that efferent Down state durations show a continuous range, but are quantized (red boxes) in the units of afferent UDS cycles, in the model and (**f**) in vivo. **g** With time measured in units of the varying afferent UDS cycles, the model (inset) predicted that the distribution of both Up and Down state durations should be multimodal, with peaks at the half integers (reflecting state transitions after an Up/Down state). In vivo data combined from all experiments showed the predicted multimodality and quantization when ECIII Up/Down state durations were measured w.r.t. variable neocortical UDS cycle lengths. The model prediction and the in vivo data matched over three orders of magnitude variation of *SPA* and *SPI* probabilities.

simulated efferent activity were extracted around each efferent Down-Up transition, sorted according to the ensuing Up state duration, and assembled into a single matrix, with each row corresponding to a single efferent Down-Up transition (Fig. 5b). The underlying afferent activity matrix for the same time points exhibited alternating bands of UDS, with integer multiples of afferent UDS fitting inside each efferent Up state (Fig. 5c). The same visualization with in vivo data matched well with model predictions (Fig. 5d). We repeated this for efferent Down states and *SPI*, yielding a similar quantitative match between the model and experiment (Fig. 5e, f). When consolidating the quantized state durations over all experiments and their matched simulations, the probability distributions for both were significantly multimodal, with peaks at half integers, indicating that ECIII state transitions were locked to the neocortical transitions, and that the ECIII skipped entire neocortical Up/Down states in integer quantities (Fig. 5g).

The multimodality of quantized durations was also observed for individual experiments and their corresponding simulations (Fig. 6a, Supplementary Fig. 14). We leveraged this distribution, unique to each cell, to investigate the precise history-dependence of *SPA* and *SPI*, further testing our model. One can imagine three scenarios. First, the *SPA* and *SPI* are entirely stochastic, in which case their probability distribution would follow a memoryless Bernoulli process, like a sequence of fair coin flips. Second, *SPA* and *SPI* arise due to some change in the overall state of the animal, such that all the *SPA* and *SPI* co-occur. However, our model predicts a third possibility: it should be rarer to have consecutive sets of *SPA* and *SPI* compared to singular events. This is because the probability of *SPA* and *SPI* is strongly history-dependent. If the network exhibits *SPA* at a given afferent Down state, the efferent network's recurrent excitation $W_{INT}$ would be more adapted than usual, reducing the resources needed to sustain

*SPA* in the next Down state, thus reducing the probability of consecutive *SPAs* (Supplementary Fig. 15). Similarly, the occurrence of *SPI* at a given afferent Up state would make the efferent network less adapted and hence reduce the probability of consecutive *SPIs*. To test this prediction, we used the first two modes of the quantized probability distribution (in Fig. 6a) to calculate $p_1$, the probability of a solitary *SPA* and *SPI*, and $p_2$, the probability that a second *SPA* and *SPI* occurred given the first (with probability $p_1$) already happened. Here, $p_2 = p_1$ for the first memoryless hypothesis, $p_2 > p_1$ for the second brain-state dependent hypothesis, and $p_1 > p_2$ for the third hypothesis, predicted by our model. The experiments strongly corroborated our predictions: the probability of *SPA* and *SPI* diminished after the first such event (Fig. 6b). The two network system thus has a "memory" of *SPA* and *SPI* due to the adaptation of the recurrent excitation $W_{INT}$ in the efferent network.

## Discussion

Persistent activity has been hypothesized to mediate working memory via reverberating activity[35,61], and has been studied extensively in vivo[62–64], in vitro[26,27,65,66], and in silica[24,25]. Its ubiquity and diversity in different cell types, brain regions, brain states, and behavior supports the hypothesis that a common mechanism could apply, and a low dimensional theory could be appropriate. Hence, we developed a mean field model to explain the recent discovery of spontaneous persistent activity in MECIII during sleep[22], as existing models focus only on stimulus evoked persistent activity during awake behavior. Using two networks of excitation-inhibition neurons and adapting excitation, with an afferent network providing excitatory input to an efferent one, our model reproduced phase locked Up-Down state (UDS) oscillations and the reported spontaneous persistent activity.

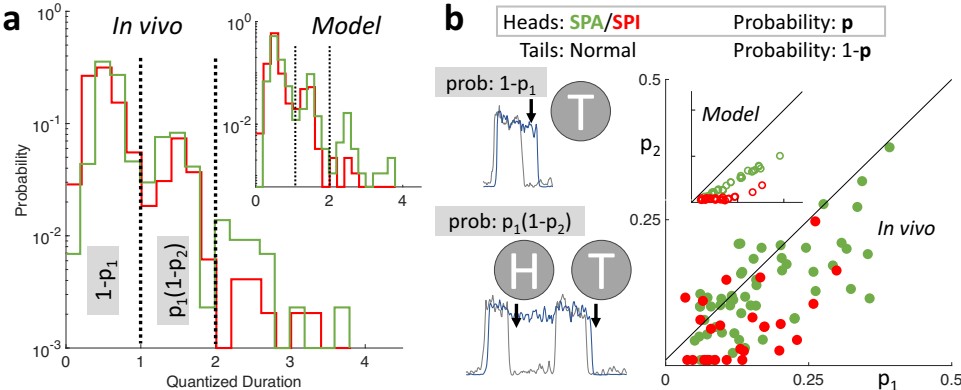

**Fig. 6 | Adaptation introduces history-dependence to SPA and SPI, a Bernoulli process. a** Quantization of SPA and SPI is evident within an individual experiment (same as Fig. 5) and its corresponding simulation (inset). The probability decreases exponentially with SPA and SPI quantized duration, reflecting a discrete Bernoulli process. **b** To find the precise history-dependence, one can imagine the efferent response to afferent UDS as a sequence of coin flips, as *SPA* and *SPI* are all-or-nothing binary events. State transitions in the afferent network destabilize the efferent network, which either persists in its current state, resulting in *SPA* or *SPI* (heads, probability *p*), or makes a corresponding transition (tails, prob. *1-p*). The

probability that a given efferent state lasts a single afferent UDS cycle is $1-p_1$ (the first mode in (**a**)), while the probability that it lasts between 1 and 2 cycles is $p_1(1-p_2)$ (the second mode in **a**), where the subscript denotes the 1st/2nd transition. Unlike a memoryless process (i.e., $p_1 = p_2$) or brain state dependent process (i.e. $p_1 < p_2$), the model (inset) predicted that the probability of *SPA* or *SPI* should decrease when conditioned on *SPA* or *SPI* having occurred previously (i.e., $p_1 > p_2$, $p < 10^{-16}$). This was confirmed in the in vivo data (main, $p < 10^{-7}$) and reflects the underlying long-term memory of the adaptation in the efferent network. Wilcoxon signed ranked test was used to show significant differences.

## Table 1 | Summary of our predictions and their experimental verification

|  | Prediction and Verification | Figures/Tables |
|---|---|---|
| **1** | Presence of SPA and SPI in EC | Figs. 1, 2, Supplementary Fig. 4 |
| **2** | MECIII has higher $W_{INT}$ than LECIII | Fig. 3e, Supplementary Tab. 2 |
| **3** | MECIII and LECIII have lower $W_{EXT}$ from neocortex compared with PAR, FRO, PRE | Fig. 3d, Supplementary Tab. 2 |
| **4** | Down-Up delay is inversely proportional to $W_{EXT}$ | Fig. 4 b, d |
| **5** | Down-Up delay has weaker dependence on $W_{INT}$ than $W_{EXT}$ | Supplementary Fig. 11a |
| **6** | Up-Down delay is proportional to $W_{INT}$ | Fig. 4b, g |
| **7** | Up-Down delay has weaker dependence on $W_{EXT}$ than $W_{INT}$ | Supplementary Fig. 11b |
| **8** | Predicted Up-Down delay vs in vivo Up-Down delay | Fig. 4e |
| **9** | Predicted Down-Up delay vs in vivo Down-Up delay | Fig. 4h |
| **10** | Distribution of SPA durations in model vs in vivo | Fig. 5a-d |
| **11** | Distributions of SPI durations in model vs in vivo | Fig. 5e-f |
| **12** | ECIII SPA and SPI are quantized by neocortical UDS | Figs. 5g, 6a, Supplementary Fig. 14 |
| **13** | SPA and SPI are history dependent Bernoulli process | Fig. 6b |
| **14** | Firing rate is nonlinear function of total excitation $W_{EXT} + W_{INT}$ | Supplementary Fig 9 |

The relevant figures/tables for each prediction are shown on the right column. Gray-colored rows indicate a quantitative match, while white-colored rows indicate a qualitative match.

Further, the model exploited the symmetry of the discrete attractor landscape to make a surprising prediction, namely the existence of persistent inactivity. In contrast to persistent activity, which involves the efferent network sustaining activity while afferent inputs have shut off, persistent inactivity involves the efferent network sustaining *inactivity* while afferent input turns on. This has not been reported before in any experimental or theoretical studies, though there are hints[67,68]. Computational studies have found coexisting Up and Down states in different neurons within the same spiking network[69,70], but these results are usually achieved when the network UDS is highly irregular and asynchronous. To test our model, we focused on the cortico-entorhinal interaction during UDS oscillations, using simultaneous LFP from the neocortex that served as a common afferent reference, along with the membrane potentials measured from anatomically identified neurons in the parietal, frontal, prefrontal, and entorhinal cortices.

The experiments confirmed the presence of both persistent activity and inactivity; we were then able to leverage these two

observables to probe the underlying network architecture and conclusively test several predictions (see Table 1 for a summary). Our framework models different brain regions by varying only two biologically relevant parameters: the strength of internal connections $W_{INT}$ within the efferent network and the strength of external input $W_{EXT}$ from the neocortex, while leaving all the other parameters unchanged. Dynamical systems analysis[55,56] showed that SPA increases with $W_{INT}$, while SPI decreases with $W_{EXT}$; thus, each cell, and the local network in which it is embedded[44], could be mapped to the $W_{INT}$-$W_{EXT}$ parameter space. Our results predicted that neocortical input onto the entorhinal region should be weaker than to other regions within the neocortex, like parietal, frontal, and prefrontal cortex. This is consistent with anatomical observations of strong intra-neocortical connections and weaker neocortical-entorhinal connections[38,39,41,71,72]. Within the entorhinal region, LECIII exhibited significantly less SPI compared to MECIII, hinting that the latter received less neocortical input. This is consistent with classic anatomical studies[73,74], which show that a higher proportion of LEC afferents originate in cortical areas compared to

MEC afferents, and more recent work[48,75] showing stronger projections from the orbitofrontal cortex, part of the prefrontal cortex, to LEC compared with MEC.

Our analysis found greater amounts of persistent activity in MECIII than in LECIII, and the model predicted that this is because the recurrent connections $W_{INT}$ should be larger within MECIII than within LECIII. This is indirectly supported by recent experiments showing greater recurrent connectivity between principal neurons within MECIII than within other MEC layers, and that MECIII network is crucial in the initiation and maintenance of the Up state during UDS in vitro in isolated EC slices[52,76]. Furthermore, excitatory cholinergic receptors are crucial for MECIII persistent activity[77], and the application of acetylcholine to MEC slice preparations in vitro causes prolonged Up states in individual cells due to increased overall excitation and more frequent and rhythmic population-wide events, consistent with our hypothesis that persistent Up states are the result of networks having increased internal excitation $W_{INT}$[78]. Finally, our results are in direct agreement with excitatory and inhibitory cell densities found by EPFL's Blue Brain Initiative[79] (Supplementary Table 2), which shows that MECIII has almost twice as many excitatory neurons per cubic millimeter compared to LECIII. Our model was not able to find a statistically significant difference between internal excitation in LECIII vs neocortical areas, and this is also supported by similar values for excitation-inhibition ratio between these areas, and a significantly higher E-I ratio in MECIII.

The model with above network connectivity not only predicted the prevalence of *SPA* and *SPI* in the efferent neurons but also predicted their relative timing to afferent neocortical activity[59], at both population-wide and single-cell resolution. Cells with higher predicted $W_{EXT}$, and thus stronger coupling, exhibited significantly shorter state transition lags, while larger recurrent excitation $W_{INT}$, and thus stronger "inertia," had longer lags, as expected. The latency patterns were quite different for Down-Up vs. Up-Down transitions: the former was more dependent on $W_{EXT}$, and the latter more on $W_{INT}$. Our results thus support the hypothesis that the Up state is terminated by internal network mechanisms but is initiated by external input[29,80]. Taken together, these resulted in systematic differences in the response latencies of MECIII and LECIII neurons during Up and Down states, which would influence the information processing in downstream hippocampal neurons[17,22] and hence the memory consolidation process via spike timing-dependent plasticity mechanisms[19].

As a direct consequence of the underlying physics of the model, we predicted that both *SPA* and *SPI* durations, while showing continuous, long-tailed distributions, should also show quantization in the units of afferent neocortical UDS cycles. This too was verified experimentally, with not just qualitative but a quantitative match between the model and experiment. Our model went further to predict that *SPA* and *SPI* were highly history-dependent, reducing the probability of consecutive *SPA* and *SPI*, and this too was confirmed in vivo. This long time-scale memory is an emergent property of the adaptation in the efferent EC network, which has been implicated in the formation and maintenance of periodic spatial firing of grid cells in MEC[81].

While persistent activity has been studied extensively as the mechanism underlying working memory, it is far more energetically expensive than persistent inactivity. Furthermore, the models involving only persistent activity have a limited storage capacity, especially when dealing with memories that require overlapping representations[82–84]. Persistent inactivity introduces a new mechanism to overcome this difficulty. From an information theoretic perspective, a 0 is just as informative as a 1. Hence, a combination of persistent activity and inactivity would be an energy and information-efficient scheme for storing overlapping memories by multiplexing the representation[85,86]. Related, our model predicted that the same neuron can show *SPA* and *SPI*, and this was experimentally confirmed. Recent theories investigated "activity-silent" mechanisms for working memory and hypothesized that the information is stored in facilitated synapses[87–89]. One prediction is that non-specific inputs can reawaken the memory ensemble after the inactive period. Our model predicts, and experiments confirm, something similar: that the efferent network is more susceptible to inputs after SPI due to falling adaptation. These dynamics between adaptation and activity could drive the production of sequences of memories in neural networks with discrete[90] and continuous phase spaces[91].

The long duration of UDS under anesthesia allowed unequivocal detection of both *SPA* and *SPI*. But, since *SPA* and *SPI* remained unchanged across a range of anesthesia depths, and *SPA* has been shown in MECIII during drug-free sleep, these results should be broadly applicable[22]. On the other hand, many biological factors that we did not consider could modulate our system wide findings. For example, in addition to the direct inputs from the parietal cortex to EC, there is substantial indirect input via the perirhinal and postrhinal cortices that we did not consider[92]. Recent studies show some cortical inhibitory neurons that remain active during the down state, which can alter the nature of cortical UDS[93]. Including these effects could help explain the discrepancy between the predicted value of the transition delays between the neocortex and EC neurons and the experimental observations. Finally, the hippocampus receives EC input and projects back to EC, and EC projects back to the frontal cortices; these connections were not included in our model, but could be studied in the future[94].

Despite this, the simple model was able to predict and match a large number of experimental observations in a quantitative, cell-by-cell manner. This may be because neural systems undergoing UDS oscillations live in a very low dimensional space, and thus our simple model has just the right complexity to describe it. The differences between brain regions and in the connectivity between brain regions could become important during other behavioral states, such as working memory. Future studies can build on our approach to study *SPA* and *SPI* in such scenarios.

Direct and indirect pathways link the entorhinal region to the hippocampus[48,94], and recent experiments have confirmed the particularly crucial role of UDS oscillations in the temporoammonic pathway, which monosynaptically projects from MECIII and LECIII to CA1, in the consolidation of newly encoded memories[95]. Further, MECIII inputs play a crucial role in driving CA1 spatial selectivity[96] during exploration, sharp-wave ripples during awake immobility[97], and hippocampal replay during rest/sleep[98]. The ripple rates are comparable to the SPA rates, and CA1 is most excited during SPA[22], suggesting that SPA could play a role in ripple generation and replay. The contribution of SPI on hippocampal activity is not yet known, but our data suggests that this would coincide with CA1 hypoactivation. The selective decoupling of medial but not lateral entorhinal activity from neocortical inputs during *SPA* and *SPI* could contribute to the selective pruning and strengthening of memory traces from the hippocampus during slow-wave sleep, thus improving the signal to noise ratio in the space of memories, thereby improving experimentally observed, post-sleep task-related performance[15]. Our model is sufficiently general and could equally apply to other networks, e.g. parietal-prefrontal network, where persistent activity is seen during working memory tasks[63,99]. Indeed, recent studies of brain activity in humans have shown that functional network connectivity during spontaneous epochs is highly dynamic[100], and that persistent activity during working memory gates the propagation of activity, and thus information, into the prefrontal network[101].

In sum, these results demonstrate that during UDS, the rich dynamics of the entire cortico-entorhinal circuit can be captured in a quantitatively precise fashion by a dynamic attractor landscape involving just two biologically important variables: the cortico-entorhinal excitation and the recurrent excitation within the entorhinal cortex. Our model is simple enough to be analytically tractable.

With just two parameters, we were able to reproduce nearly a dozen different experimental observations in a quantitatively precise fashion. This strong, quantitative match between a simple model and in vivo data thus yields a general theory of interaction between large networks during spontaneous activity and reveals several novel emergent properties such as spontaneous persistent activity as well as inactivity that provides energy efficient memory and that could explain 'activity silent' working memory[82,86,92]. Further, this reveals the nature of cortico-entorhinal functional connectivity during slow oscillations in vivo, and the differential nature of this connectivity between MECIII vs LECIII. Given the strong, quantitative match between theory and data, we hypothesize that similar differential functional connectivity will hold in other brain states too, such as active exploration. This approach provides a powerful technique to understand the functional connectivity between large networks of neurons in vivo.

## Methods

### A single E/I Mean field model

We constructed a, mean-field network[53,54] that can support UDS oscillations[29–31,33], which modeled the average activity of two populations of neurons: one excitatory and one inhibitory (Supplementary Fig. 1). This average activity level is a dimensionless number whose value ranged between 0 and 1. Since UDS are slow and synchronous oscillations, this level of granularity is sufficient and one does not need to include faster variables like spikes. This greatly reduces the number of free parameters and keeps the equations of motion analytically tractable, with high predictive power. The time evolution of the average excitatory $E(t)$ and inhibitory $I(t)$ activity is governed by:

$$\tau_E \frac{dE}{dt} = -E + \Omega_E(W_{EE}E - W_{EI}I - W_{EA}A + \xi + i_E) \quad (1)$$

$$\tau_I \frac{dI}{dt} = -I + \Omega_I(W_{IE}E - W_{II}I + \xi) \quad (2)$$

Where $\tau_{E/I}$ is the time constant of each network ($\tau_E = 10$ ms, $\tau_I = 5$ ms), consistent with experimental data on the membrane time constants. We first consider the case where there is no external input, so $i_E = 0$. The $\Omega_{E/I}$ response function is a standard threshold-linear function with saturation:

$$\Omega_{E/I}(x) = \begin{cases} 0 & \text{if } x < \theta \\ g_{E/I}(x - \Theta_{E/I}) & \text{if } \theta < x < \theta + 1/g_{E/I} \\ 1 & \text{if } x > \theta + 1/g_{E/I} \end{cases} \quad (3)$$

where $g_{E/I}$ is the slope of the input-output relationship for each neural population ($g_E = 6$, $g_I = 30$) and $\theta_{E/I}$ is the threshold input needed for each population ($\theta_E = 0.0517$, $\theta_I = 0.2778$). For both populations, the inputs are simply the sum of currents from neural populations in the network, given by synaptic weight $W_{XY}$, from population $Y$ to population $X$, multiplied by the source activity $E/I$ ($W_{EE} = 1$, $W_{II} = 0.083$, $W_{EI} = 0.166$, $W_{IE} = 1.66$). There is additional noise current $\xi$, drawn from a gaussian distribution (mean of 0, std. 0.03) to simulate random fluctuations within the network activity. In all simulations and theoretical analysis, the input remained in the linear regime and never reached saturation. The excitatory population has an additional term to describe its internal, activity-dependent adaptation $A$, with weight $W_{EA} = 0.166$:

$$\tau_A \frac{dA}{dt} = -A + W_{AE}E \quad (4)$$

Where the time constant $\tau_A = 300\,ms$ is much larger than the time constants of excitation and inhibition, and the modulation due to excitation is $W_{AE} = 1.1$. This time constant is in line with most mean field models of UDS, and is thought to arise in the dendrites[102]. $E = 0, I = 0$ is a steady state of this network, and corresponds to the Down state observed during UDS. Since the adaptation parameter is so slow-varying, we can consider a snapshot of the network at a fixed adaptation $A^*$ and consider the state space of all possible realizations of activity $E/I$. We can solve for the nullclines of each population by setting the derivative of the activity in each population to zero, and solving for $E$:

$$E = \frac{g_E W_{EI}I + g_E\left(W_{EA}A^* + \Theta_E\right)}{g_E W_{EE} - 1} \quad (5)$$

$$E = \frac{(1 + g_I W_{II})I + g_I \Theta_I}{g_I W_{IE}} \quad (6)$$

These are plotted in Fig. 1b and Supplementary Figs. 1–3. In order to have a stable Up state, these two nullclines must intersect at non-zero values for $E$ and $I$, which is possible under the condition that

$$\Theta_I > \frac{g_E W_{IE}}{g_E W_{EE} - g_E W_{EA}W_{AE} - 1}\Theta_E \quad (7)$$

$$g_E < \frac{1 + g_I W_{II}}{W_{EE}(1 + g_I W_{II}) - g_I W_{EI}W_{IE}} \quad (8)$$

These conditions are satisfied by our choice of parameters, and ensure that the excitation nullcline is steeper than the inhibition nullcline but has a smaller $E$-intercept, thus ensuring an intersection. We identify this intersection with the neurological Up state, where both excitatory and inhibitory populations exhibit sustained firing. Its coordinates are

$$E = \frac{W_{EI}\Theta_I - \left[W_{II} + \frac{1}{g_I}\right]*\Theta_E}{W_{EI}W_{IE} - \left[W_{EE} - \frac{1}{g_E} - W_{EA}W_{AE}\right]*\left[W_{II} + \frac{1}{g_I}\right]} \quad (9)$$

$$I = \frac{\left[W_{EE} - \frac{1}{g_E} - W_{EA}W_{AE}\right]*\Theta_I - W_{IE}\Theta_E}{W_{EI}W_{IE} - \left[W_{EE} - \frac{1}{g_E} - W_{EA}W_{AE}\right]*\left[W_{II} + \frac{1}{g_I}\right]} \quad (10)$$

A third fixed point is found at $E = 0$ and $I = \frac{g_E(W_{EA}A^* + \Theta_E)}{g_E W_{EE} - 1}$. This point is unstable and lies on the separatrix, which marks the boundary between two regions of stability. Thus, one can imagine that the network sits in a potential landscape with two minima and one energy barrier in the middle[103].

The local stability of the Up state can be found by linearizing the differential Eqs. 1–2 about the Up state fixed point, and ensuring that the eigenvalues of the matrix of coefficients have a negative real part, signifying that fluctuations will exponentially decrease[30,56]. For our 2D matrix, this is equivalent to imposing that the determinant of the coefficients matrix is positive and the trace is negative. These conditions yield the following relations between connectivity, time-scale, and gain:

$$\left[W_{II} + \frac{1}{g_I}\right]*\left[W_{EE} + \frac{1}{g_E}\right] < W_{EI}*W_{IE} \quad (11)$$

$$\tau_I * (g_E W_{EE} + 1) < \tau_E * (g_I W_{II} + 1) \quad (12)$$

These conditions are satisfied by our choice of parameters.

The global stability of each fixed point is inversely related to the distance of the fixed point from the unstable separatrix. The closer each stable fixed point (the Up or Down state) is to the separatrix, the

less relatively stable that fixed point becomes, since random noise has a higher chance of kicking the network over the boundary. Notably, the variable $A^*$ is simply an additive constant to the excitation nullcline, the dynamics of which determines the positions of intersection for both the stable Up state as well as the separatrix. As the network remains in the Up state, the adaptation variable increases, effectively shifting the excitation nullcline up. This not only decreases the overall firing rate in the Up state by shifting the fixed point, it also stabilizes the Up state by bringing the separatrix closer to the Up state. A kick from random noise eventually forces the network to transition into the down state. Here, adaptation recovers back to zero, shifting the excitation nullcline down, thereby bringing the separatrix closer to the Down state fixed point; eventually, a noisy kick forces the network into the Up state, where the cycle repeats.

### Coupled networks: persistent activity/inactivity

To model spontaneous persistent activity, we consider two identical networks, each described by the above equations. Further, the afferent network provides a weak excitatory input $W_{EXT}$ into the excitatory population of the efferent network. We use $W_{EXT}$ to refer to this synaptic weight from the afferent to the efferent network and $W_{INT}$ to refer to the internal excitatory-excitatory weight in the efferent network ($W_{EE}$ in Eqs. 1–12). Effectively, this means that for the efferent network there is an external input into the excitatory population $i_E(t) = W_{EXT} \cdot E_A(t)$, where $E_A(t)$ is the activity of the excitatory population in the afferent network. If the connection strength $W_{EXT}$ between the afferent and efferent excitatory populations is sufficiently strong, the two networks UDS oscillations phase lock, as the transitions between states in the efferent network are no longer due to independent noise but the timed increase and decrease in input coming from the afferent network. Similar to previous results, the connection strength $W_{EXT}$ must be about an order of magnitude smaller than the internal connections $W_{INT}$ in order to show desynchronization[30].

What happens when both networks are in the Up state and the afferent input transitions into the Down state? This cuts off afferent input $W_{EXT}$, immediately shifting the efferent excitation nullcline to the left, thereby destabilizing the efferent Up state. The efferent network can either remain in the Up state through its own recurrent excitation or follow the afferent and transition into the Down state. Instances when the efferent remains in the Up state are termed "spontaneous persistent activity (SPA)." It follows from the stability arguments outlined earlier that by increasing the distance between the Up state fixed point and the separatrix, one can increase the stability of the Up state, thereby increasing the probability that the network will display SPA. If we refer to the excitation nullcline equation [Eq. 5], we see that increasing $W_{INT}$ ($W_{EE}$ in Eqs. 1–12) results in the downward scaling of the nullcline: the Up state fixed point shifts to the right, and the unstable fixed point shifts downward. Combined, this has the effect of increasing the stability of the Up state, thus leading to higher probability of SPA.

The converse scenario applies for the Down state, where both networks are in the Down state and the afferent transitions into the Up state. This suddenly increases the input, shifting the excitation nullcline down, destabilizing the Down state. The efferent network can either follow into the Up state or remain in the Down state; the latter case we term "spontaneous persistent inactivity" (SPI). The size of downward shift due to the incoming current from the $W_{EXT}$ synapse has a direct consequence on the stability of the Down state: the larger the weight, the more the shift, and thus the more destabilized the DOWN state becomes. Thus, decreasing the synaptic weight $W_{EXT}$ increases the probability of SPI. Indeed, simulations where we modulated both $W_{INT}$ and $W_{EXT}$ confirmed our hypothesis on the dependence of persistent activity and inactivity on these two variables (Fig. 1b). All simulations were performed using MATLAB on the UCLA Hoffman2 Computing Cluster with time-step 0.2 ms using the Runge-Kutta method.

### Animals, surgery, and histology

Methods were similar to those described previously[22]. Briefly, data were obtained from 136 C57BL6 male and female mice aged postnatal day (p) 25-43 (p32 ± 1) weighing 12–21 g (17.5 ± 0.4 g). All mice were housed under a 12 h light, 12 h dark cycle (7AM to 7PM light) at 23 ± 2 °C and a relative humidity of 55 ± 5% with *ad-libitum* access to food and water. Mice were anesthetized with urethane (1.64 ± 0.03 g urethane / kg body weight intraperitoneal). Body temperature was maintained at 37 °C with the help of a heating blanket. The animals were head-fixed in a stereotaxic apparatus and the skulls exposed. A metal plate was attached to the skull and a chamber formed with dental acrylic, which was filled with warm cerebrospinal fluid. Two 1 mm diameter holes, one for the LFP recordings and one for the whole-cell recordings, were drilled over the left hemisphere and the underlying dura mater was removed.

After electrophysiological recordings, mice were euthanized by transcardial perfusion with 0.1 M phosphate buffer, followed by 4% Paraformaldehyde solution, and 150–200 μm thick brain sections were processed with the avidin-biotin-peroxidase method. Sometimes, a subsequent Nissl stain was applied before embedding. Visualization of biocytin filled neurons allowed for the determination of cell type and recording site. Unidentified neurons were excluded from analysis. All experimental procedures were carried out according to the animal welfare guidelines of the Max-Plank-Society.

### Electrophysiology and data acquisition

Local field potentials (LFPs) were recorded with an 8 site single-shank multisite probe (NeuroNexus Technologies). LFP from layer 2/3 of posterior parietal cortex (2 mm posterior to bregma, 1.5 mm lateral) was used to characterize neocortical Up-Down states. In vivo intracellular membrane potential ($V_m$) was recorded in whole-cell configuration by using borosilicate glass patch pipettes with DC resistances of 4–8 MΩ and filled with a solution containing 135 mM Potassium Gluconate, 10 mM HEPES, 4 mM Potassium Chloride, 10 mM Phosphocreatine, 4 mM MgATP, 0.3 mM $Na_3GTP$ (adjusted to pH 7.2 with KOH), and 0.2% biocytin for subsequent histological identification. Whole-cell recording configuration was achieved as described previously[104]. Relative to bregma, the anteroposterior (AP), mediolateral (ML) and dorsoventral (DV) coordinates of the craniotomies for the $V_m$ recordings were made around −4.5 mm AP and 4 mm ML for MEC; −3.5 to −4 mm AP, 4.5 mm ML and 4 mm DV for LEC; −1.5 to −2 mm AP and 1 mm ML for parietal cortex; 1 to 1.5 mm AP and 1 mm ML for frontal cortex; 2 to 3 mm AP and 0.5 to 1 mm ML for prefrontal cortex.

The average initial series resistance was 46 MΩ, and $V_m$ values were corrected for the estimated junction potential of approximately +7 mV.

The $V_m$ was acquired by Axoclamp-2B (Axon Instruments) and fed into a Lynx-8 amplifier (Neuralynx). The $V_m$ and LFP were recorded by an HS16 preamplifier (Neuralynx) for about 20–40 minutes. The complete recording was used for subsequent statistical analysis. The LFP were sampled at 2 kHz, low-pass filtered below 475 Hz, and amplified 1000–5000 times. The membrane potential was low-pass filtered below 9 kHz, sampled at 32 kHz, and amplified 50–150 times. Simultaneously, the DC value of $V_m$ was recorded by an ITC18 interface (Instrutech) under the control of Pulse software (Heka) or by a Micro1401 with Spike2 software (CED). Some of these DC-coupled data were recorded in discontinuous sweeps of 7 or 10 s, separated by 5 or 2 s, respectively. Data from previous work[22] was supplemented with additional recordings from MECIII and LECIII. The parietal, frontal, and prefrontal $V_m$ measurements are entirely new.

### Data preprocessing

All analysis was restricted to subthreshold fluctuations in the membrane potential by removing spikes as follows. The temporal derivative of the bandpass-filtered (100 Hz–8 kHz) membrane potential signal

was computed, and times when this derivative exceeded 10 standard deviations above the mean were taken as spike times. Spike waveforms were then removed by replacing 3 ms of data following the onset of each spike by linear interpolation of adjacent values. To remove the 50 Hz mains hum and its many harmonics, 8-pole bandstop filters were used at 45–55 Hz, 95–105 Hz, 145–155 Hz, 195–205 Hz, 245–255 Hz, and 295–305 Hz.

### Explicit-duration Hidden Markov model detection of Up and Down states

Synchronized epochs, wherein the LFP and cortical $V_m$ undergoes synchronous transitions of Up and Down states (UDS), were selected by locating and eliminating periods of data with desynchronized activity where UDS are absent. Previously outlined methods[58] were closely followed. Briefly, the spectrogram of the signal was computed in 15 s overlapping windows using multi-taper methods (Chronux Matlab toolbox) with a time-bandwidth product of 4, and seven tapers. The maximum log power in the range of 0.05–2 Hz and the integral of the log power in the 4–40 Hz range we then used to locate and remove desynchronized epochs in the data.

The remaining data exhibited UDS. UDS of both membrane potential and neocortical LFP were classified using two state explicit-duration hidden Markov models (EDHMMs). The $V_m$ and LFP were first filtered in the low frequency (0.05–2 Hz) range, and a gaussian observation EDHMM was fit to the filtered signal, with inverse Gaussian models of the state duration distributions. The means of the state-conditional gaussians were slowly varying functions of time, where the parameters were estimated over a 50 s window length. We found the maximum likelihood parameter estimates of the EDHMM, and computed the resulting "Viterbi" sequence, which was used to define UDS oscillations.

### Assignment of corresponding neocortical-entorhinal state transitions

Given two UDS sequences, one for the neocortical LFP (the afferent network in the simulation) and one for the entorhinal $V_m$ (efferent network), the fine temporal relationships and quantized duration was calculated by first assigning each Up/Down state in the $V_m$ to its corresponding set of trigger states in the LFP. This was done through a greedy search algorithm, where in each iteration of the algorithm, the Up/Down state initiations were linked to the closest Up/Down state initiations in the corresponding LFP. Note that this does not guarantee a one-to-one mapping from $V_m$ states to LFP states; those $V_m$ states which map onto more than one LFP state are termed "persistent." The quantized duration of a $V_m$ state was calculated as the number of total LFP states (both Up and Down) that would fit inside a particular $V_m$ state, with each Up and Down state in the cycle contributing to 0.5 units of time (Supplementary Fig 10).

### Model fitting

The $W_{EXT}$-$W_{INT}$ parameter space was divided into a 100-100 grid, and each point was taken as the input into 5 independent simulations of length 1000 s. For each experiment, we calculated the "distance" between the experimental data SPA/SPI level and each simulated SPA/SPI level (Supplementary Fig. 7). Let $\phi_{SPA/SPI}$ denote the proportion of efferent states which were classified as SPA/SPI (see above) in the experiment, and $\xi_{SPA/SPI}$ denote the proportion in a given simulation. The distance between the experiment and any particular simulation is then given by $d = \sqrt{(\phi_{SPA} - \xi_{SPA})^2 + (\phi_{SPI} - \xi_{SPI})^2}$. The simulation with the minimum distance to the experiment was chosen as the best fit (Fig. 3a). One could also take $\phi$ and $\xi$ to denote the proportion of afferent LFP states which were classified as 'skipped' by the efferent network, and use this for the distance metric. Results of this fit are shown in Supplementary Fig. 8, and are virtually identical to the procedure used in Fig. 3a of the main text.

### Statistics and hypothesis testing

Central tendencies and variability is reported as mean plus/minus standard deviation, unless otherwise noted. All hypothesis tests were performed using two-sided nonparametric Wilcoxon rank-sum tests for equal medians. Wilcoxon signed rank tests were used for paired comparisons or one-sample tests. Correlations were computed using Spearman's rank correlation coefficient. A $p$-value of less than 0.05 was used for statistical significance.

### Reporting summary

Further information on research design is available in the Nature Portfolio Reporting Summary linked to this article.

## Data availability

Data is available upon request, please contact the corresponding author. Source data are provided with this paper.

## Code availability

Code to simulate the coupled network is available on github.com/krishnizzle/Coupled_UDS_Networks.

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

## Acknowledgements

We thank J. Moore for comments on the manuscript. James McFarland first noticed SPI. This work used computational and storage services associated with the Hoffman2 Shared Cluster provided by UCLA Institute for Digital Research and Education's Research Technology Group. This work was funded by the W.M. Keck Foundation, an AT&T research grant, NSF grant #1550678, NIH grant #1U01MH115746, and NIH-BMBF-CRCNS grant #5R01MH092925-02, all to MRM. Preliminary findings were presented in Society for Neuroscience meetings (2017-2019).

## Author contributions

K.C. did theory, simulations, and data analysis. S.B. and T.G.H. did experiments. M.R.M. participated and supervised all aspects. J.M. first noticed SPI and did preliminary analysis. K.C. and M.R.M. wrote the paper, with input from S.B.

## Competing interests

The authors declare no competing interests.
