## [Peer Review File · Nature Communications]

Spontaneous persistent activity and inactivity in vivo reveals differential cortico-entorhinal functional connectivityREVIEWER COMMENTS

Reviewer #1 (Remarks to the Author):

This is a very nice study of the relationship between up and down states across multiple areas. The experimental data is modeled with a simple, intuitively understandable model, which has a low number of parameters, and which leads to several important, nontrivial predictions that are borne out in the data. This work tightly integrates experiment and theory, should be of broad interest, and should provide especially valuable insights to the many researchers interested in network states and cortical interactions. I only have a few clarification comments and questions.

1. On p6 the authors write: "In contrast to Down-Up transitions, the latency of the efferent Up-Down transition should be relatively insensitive to WEXT compared to WINT. This prediction too was supported across individual neurons within MECIII, within LECIII, and across the MECIII vs LECIII ensemble (Sup. Fig. 8)."

Wouldn't a larger WEXT mean that when the afferent input area drops down from UP to DOWN that the efferent area will see a larger drop in input, which might be expected to help push the efferent area into the DOWN state more quickly? Does the model show this to be the case, for example in a phase plot similar to Fig1B but considering latency rather than SPI % and SPA % (it does seem to be the case in the example in Fig3D)?

2. On p9, the authors write: "Similarly, the occurrence of SPI at a given afferent Up state would make the efferent network less adapted and hence reduce the probability of consecutive SPIs."

Is the time scale of adaptation long enough to have an effect across a single up and down state? The adaptation time constant is apparently set to 300 ms in their model, but the Up and Down states appear to generally be approximately 500 ms to 1 second long. This means it takes 1-2 seconds for a cycle until the next opportunity for an SPI, significantly longer than the 300 ms time constant. How was the 300 ms time constant chosen? Can the effect of adaptation be tested for its effect on consecutive SPI probability in the model?

Minor points:

1. On p7, the authors write: "Briefly, our model implies that neocortical input into MECIII is weaker than into LECIII, and still weaker than into other neocortical regions, like parietal, frontal, and prefrontal cortices."

As the parietal cortex is where the LFP recordings are made, it seems expected that inputs from parietal to EC would be weaker than from parietal to itself. Also, the model assumption of only a unidirectional connection from the LFP area to the intracellularly recorded area would seem to be much less probable within-area. Thus it may make sense to remove "parietal" here, unless it is argued that the LFP and intracellular recording sites are distant enough to be in different areas.

Reviewer #2 (Remarks to the Author):

Persistent neuronal activity describes the phenomenon of continued neural firing after the triggering input has been removed. Recently, a novel form of persistent activity has been discovered in the hippocampus of the mouse during Up-Down states that is triggered spontaneously (thus termed 'spontaneous persistent activity'; SPA), but its underlying neuronal mechanisms have remained unclear (Hahn et al., 2012, Nat Neurosc). In their manuscript, Choudhary and colleagues propose a theory underlying the generation of SPA. The proposed model expands previous models by adding long-range interactions between an afferent and an efferent module/node. Two key parameters are being varied: the strength of the (unidirectional and excitatory) long-range coupling between the nodes (W_{ext}) and the strength of recurrent excitation (W_{int}). Despite its simplicity, the model makes a non-trivial and novel prediction: the existence of spontaneous persistent inactivity (SPI), i.e., activity-silent periods in the efferent module despite activity in the afferent module. The authors report convincing evidence for the existence of SPI also in vivo. Having demonstrated the existence of both SPA and SPI in the hippocampus (but also parietal cortex), the authors simulate their model for a range of parameter combinations, through which they demonstrate that SPA increases with W_{int} , while SPI decreases with W_{ext} . Moreover, they fit the model to individual neuron's SPI/SPA fractions, resulting in estimations of W_{int} and W_{ext} for each neuron. This allowed them to derive and test a series of specific predictions in vivo in order to further validate their model. In sum, the authors provide a plausible neuronal mechanism for the generation of SPA and SPI in parts of the hippocampus (and the absence thereof in others), based on the interaction between long-range (cortico-hippocampal) and recurrent excitation.

This is an interesting paper and I enjoyed reading it. It is well-structured, well-argued and it tackles a timely topic. The authors do a good job combining theory/simulation and experiment. The computational mechanism that is being proposed is simple and elegant. Despite the simplicity of the

proposed mechanism, the model makes a series of non-trivial predictions (e.g. the existence of SPI and the quantized nature of SPI/SPA) and the authors do a good job testing those predictions in vivo, thus lending validity to their model. The results are of relevance to a wide range of neuroscientists that work on the neural substrates of information retention during working memory as well as information integration during decision-making.

Below are a few comments, in no particular order:

1) The delay between the membrane potential fluctuations of neurons recorded in parietal cortex and the local field potential recorded from the same region is surprising. Some negative delay could of course happen just by chance, simply by virtue of sampling only a subset of neurons in the area, but the Vm fluctuations seem to precede the LFP by roughly 100 ms (quite consistently across the recorded neurons), which appears to be quite long. As the authors assume that the LFP provides the afferent (input) signal, I wonder to what extent the negative delays invalidate fitting the model to the SPI/SPA data obtained from parietal cortex neurons? I understand that the fitting procedure is agnostic with respect to the precise timing of the signals (as it only uses SPI/SPA fractions to obtain the best parameter combination), but in my perspective, there remains some conceptual mismatch here, as the afferent signal shouldn't lag behind the efferent signal.

2) Related to the previous point: what are the delays between the LFP and the neurons in frontal cortex? Do the authors also observe negative delays here (i.e., Vm fluctuations preceding the LFP)?

3) The model the authors propose is based on the Wilson-Cowan equations. In a recent paper [*], it was shown that the WC model does not accurately predict neuronal population responses to input pulses to excitatory and inhibitory populations, thus, the WC equations don't model cortex accurately. Do the authors think this could pose a problem for the application of the model in the present case?

[*] Lin, Okun, Carandini, & Harris (2020) Equations governing dynamics of excitation and inhibition in the mouse corticothalamic network. bioRxiv. doi: <https://doi.org/10.1101/2020.06.03.132688>

4) The fraction of SPA/SPI seems to be largely identical for neurons in parietal and the neurons in frontal cortex and, as a result, the fitted model parameters are largely similar (based on the results obtained from the alternative fitting procedure shown in Supplementary Fig. 7). Considering that many differences have been reported between microcircuits in parietal and frontal cortical regions with respect to the level of recurrent excitation, the level of inhibition, as well as anatomical connectivity (e.g. Fulcher et al., 2019, PNAS; Wang, 2020, Nat Rev Neurosci), I am a bit surprised that the fitted model parameters (and the resulting dynamics) are so similar. Is this something to be expected or does it reflect the inability of the model to generalize beyond cortical-hippocampal interactions?

5) Somewhat related to the previous point, the model predicts that neurons with greater net excitatory input ($W_{ext} + W_{int}$) show higher firing rates, as confirmed by the recordings (with MECIII neurons showing higher firing rates than LECIII neurons). How do the firing rates look for the neurons in parietal and frontal cortex and how do they relate to the firing rates predicted by the model (based on the estimated net excitatory input)? As the authors state in their discussion that the "(...) model is sufficiently general and could equally apply to other networks, e.g. parietal-prefrontal network (...)", it would be important to see if the model makes accurate predictions for those areas. If not, this (and possible reasons for it) should be discussed.

6) On page 6, last paragraph, the authors write: "These latencies were more correlated with the predicted W_{int} and W_{ext} values than with simply the levels of SPA or SPI (Sup. Fig. 8), further supporting the model". Looking at the figures, the differences in correlation values are not very pronounced. It seems to be that a statement like this requires a statistical comparison of the obtained correlation values.

Others

- Supplementary Figure 1: There seems to be a mistake in the figure caption related to panel A, where it says that the adaptation time constant is 500 s.

Thank you for giving us the opportunity to revise our manuscript. We appreciate the reviewer's constructive suggestions. Addressing these took considerable time and effort but the outcome was worth it. The revised manuscript has several new results that make the theory-experiment match even stronger. Major new results and data are contained in 1 new main figure, 1 new main table, 4 new supplementary figures, and 2 new supplementary tables.

Importantly, the supplementary table shows a quantitative match between theory and in vivo experiments, as well as publicly available connectomics data, even though the theory is very simple and has only two free parameters. We believe this level of quantitative match between theory and in vivo experiments is unprecedented. It provides a fundamentally new understanding of neural circuit dynamics in vivo. It demonstrates a new approach, of physics style simple theory that can be tested quantitatively on in vivo data to reveal the functional connectome.

Major results are:

1. A major change to the manuscript is the treatment of temporal predictions made by our theory. In the previously submitted version, we showed a qualitative match between predictions made by the theory and experiments. In this revision, we did additional simulations and developed analysis techniques to directly test our theory. We made several *quantitative* predictions about the timing of neural responses as a function of connectivity parameters, all were confirmed by the analysis of experimental data. We believe provides further strong support for our theory, which can explain a myriad of phenomena despite its simplicity and elegance. This has some surprising new results listed below.
 - a) SPA probability depends exponentially on W_{EXT} and SPI probability depends exponentially on W_{INT} . The exponent is universal across all the brain regions and explains a large fraction of variance spanning 3 orders of magnitude. This is very surprising since in the experimental data other parameters would vary across neurons and brain regions, e.g. depth of anesthesia, brain region, specific inputs to the neuron etc. Yet, just the two parameters not only capture these results, but are universal across brain regions. This suggests that the other variables do not play as big a role as these two variables we have identified. The dependence of SPA on W_{INT} (or SPI on W_{EXT}) is weaker but still quite significant and also exponential as captured by the model.
 - b) The latency of Down-to-Up transition between two brain regions follows exactly the pattern predicted by the model across all brain regions. The Down-to-Up latency shows a linear dependence on W_{EXT} spanning 500ms. The Up-to-Down latency shows a linear dependence on W_{INT} spanning 1100ms.
 - c) We then directly compared the Down-to-Up latency in the measured in the experimental data and in simulations, these two matched remarkably well across brain regions. This was repeated for Up-to-Down latency, with similar results.
 - d) We simulated the firing rates of neurons in different brain regions using our model. The predicted firing rates were strongly correlated with the experimentally observed results. This revealed a nonlinear dependence of firing rates on the net excitatory drive received by each neuron.

These results provide surprising and a unified theory of interaction between diverse networks during Up-Down state oscillations to reveal the underlying functional connectivity.

2. New results from anatomical connectivity data across brain regions that match remarkably well with the model predictions of recurrent versus feed-forward connection strength across brain regions (Supp. Table 2). This further supports the theory, highlighting the ability of this technique to estimate differential recurrent vs. feed-forward connectivity between two networks using statistics gathered from electrophysiological data.
3. Separation of frontal and prefrontal data in all figures.

Following are the point-by-point responses (in blue) to reviewer comments (black).

Reviewer #1:

1. On p6 the authors write: “In contrast to Down-Up transitions, the latency of the efferent Up-Down transition should be relatively insensitive to W_{EXT} compared to W_{INT} . This prediction too was supported across individual neurons within MECIII, within LECIII, and across the MECIII vs LECIII ensemble (Sup. Fig. 8).”

Wouldn't a larger W_{EXT} mean that when the afferent input area drops down from UP to DOWN that the efferent area will see a larger drop in input, which might be expected to help push the efferent area into the DOWN state more quickly?

We agree with your reasoning, that a larger W_{EXT} would mean that the efferent network should transition to the DOWN state more quickly, and this effect is shown in the revised Supplementary Figure 11Bii. In our original text, we were simply asserting that the effect of W_{EXT} on the UP-DOWN transition should be weaker than on the DOWN-UP transition. We have emphasized and clarified this point in the revised text.

Does the model show this to be the case, for example in a phase plot similar to Fig1B but considering latency rather than SPI % and SPA % (it does seem to be the case in the example in Fig3D)?

This is an interesting point. To address this, we did additional simulations, this time keeping track of transition latencies. The latencies from the simulation are included in a new figure: main Fig. 4B. In accordance with our qualitative predictions, the DOWN-UP transition delay is primarily controlled by W_{EXT} , and the UP-DOWN transition delay is primarily controlled by W_{INT} . But as you reasoned, the UP-DOWN delay is weakly modulated by W_{EXT} as well, as increasing W_{EXT} reduces the transition delay.

This new analysis presented another opportunity to *quantitatively* test our model against experiment, as we now have two new observables (the UP-DOWN and DOWN-UP transition latencies). This analysis is presented in the new Fig. 4E/H and Supp. Fig 11. Specifically, Fig. 4E (4H) shows that the DOWN-UP (UP-DOWN) delay observed in the experiment is highly correlated with the predicted delay from the fitted model. This independent test of model on data greatly strengthens the quantitative match between theory and experiment.

This procedure also generated some surprises about the precise value of the delays (which are beyond the scope of our model). Please see revised manuscript Supp. Fig 12.

2. On p9, the authors write: "Similarly, the occurrence of SPI at a given afferent Up state would make the efferent network less adapted and hence reduce the probability of consecutive SPIs."

Is the time scale of adaptation long enough to have an effect across a single up and down state? The adaptation time constant is apparently set to 300 ms in their model, but the Up and Down states appear to generally be approximately 500 ms to 1 second long. This means it takes 1-2 seconds for a cycle until the next opportunity for an SPI, significantly longer than the 300 ms time constant. How was the 300 ms time constant chosen? Can the effect of adaptation be tested for its effect on consecutive SPI probability in the model?

The time scale of adaptation was chosen based on previous mean field models of UP-DOWN oscillations, which used this value to obtain a quantitative match between theory and experiment. We hypothesize that this slow time constant arises in the dendrites, perhaps via calcium channels leading to long-duration fluctuations reported recently (Moore et al. 2017) also known as plateau potentials. This is clarified in the text.

In these models, it is common for UP and DOWN states to last longer than this time scale, because this time scale simply represents approximately how long the adaptation changes by a factor of e , not how long it takes for the neuron to fully recover from adaptation and jump to the next transition. In addition to adaptation, transitions between states are strongly influenced by stochastic noise in our model, and this noise is sufficient to induce a transition only when the adaptation has reached asymptotic values at the two extremes, which can take several time constants to achieve.

Changing the time-scale or the strength of adaptation will require a change in many other parameters, e.g. the strength of recurrent inhibition, to keep the network in an E-I balanced state, causing a parameter explosion. As a result, this will also cause a major change in the basic characteristics of the underlying UP-DOWN oscillation itself (see Ghorbani et. al., Reference 30) making it difficult to directly test its effect on higher-order phenomena like SPA and SPI. To avoid these challenges but still test the effect of adaptation on consecutive SPA and SPI probabilities, we instead conducted a comparative analysis of the magnitude of adaptation during SPA states versus non-SPA, synchronous UP states, and similarly between SPI states and non-SPI, synchronous DOWN states (Supp. Fig 15).

The model predictions were further corroborated by this analysis. Adaptation grows during UP states until a Down transition is induced, upon which adaptation decreases during the DOWN state. During SPA states, however, the adaptation in the efferent network remains near-saturating high levels (Supp. Fig 15A) despite the afferent network going to the DOWN state. In the following UP state, the efferent network's adaptation grows again, and thus achieves an even higher value. This reduces the magnitude of noise needed to terminate the UP state, thus reducing the probability of a second subsequent SPA state. In quantitative terms, the efferent network achieves the maximum level of adaptation during SPA states (Supp Fig 15B) and thus is much more likely to transition to the DOWN state when the afferent input is terminated. Further, during SPA states, the adaptation is always higher during the second half than the first half (Supp Fig 15C). This directly leads to the efferent network being less likely to continue in the SPA state, and thus the probability of consecutive SPA states is diminished. A similar analysis with SPI states shows why consecutive SPI probability is diminished (Supp Fig 15D-F).

Minor points:

1. On p7, the authors write: “Briefly, our model implies that neocortical input into MECIII is weaker than into LECIII, and still weaker than into other neocortical regions, like parietal, frontal, and prefrontal cortices.” As the parietal cortex is where the LFP recordings are made, it seems expected that inputs from parietal to EC would be weaker than from parietal to itself. Also, the model assumption of only a unidirectional connection from the LFP area to the intracellularly recorded area would seem to be much less probable within-area. Thus, it may make sense to remove “parietal” here, unless it is argued that the LFP and intracellular recording sites are distant enough to be in different areas.

There was a significant distance (more than 300um) between the site of LFP and MP recordings within parietal cortex. The patch pipette has a large diameter at the base and so does the silicone probe manipulator. On the other hand, we previously showed that UDS properties are synchronized across large distances (Petersen et al. 2004). So, a detailed explanation of whether the parietal cortical data should/not be included in these results is nontrivial. We would like to keep these results as they are.

Reviewer #2:

1) The delay between the membrane potential fluctuations of neurons recorded in parietal cortex and the local field potential recorded from the same region is surprising. Some negative delay could of course happen just by chance, simply by virtue of sampling only a subset of neurons in the area, but the Vm fluctuations seem to precede the LFP by roughly 100 ms (quite consistently across the recorded neurons), which appears to be quite long. As the authors assume that the LFP provides the afferent (input) signal, I wonder to what extent the negative delays invalidate fitting the model to the SPI/SPA data obtained from parietal cortex neurons? I understand that the fitting procedure is agnostic with respect to the precise timing of the signals (as it only uses SPI/SPA fractions to obtain the best parameter combination), but in my perspective, there remains some conceptual mismatch here, as the afferent signal shouldn't lag behind the efferent signal.

This is an interesting point. The delay between the LFP and membrane potential within the parietal cortex is negative, i.e. the membrane potential fluctuations appear before those in the LFP by 82 ms for all neurons (Supp. Table 1). The causes for this ‘negative delay’ remain to be determined. We hypothesize that this is because the LFP is the result of EPSPs generated in the neural activity within a brain region and not the net afferent input to that brain region, as commonly assumed. This hypothesis is supported by a strong prediction of the model that that W_{EXT} is nearly ten times smaller than W_{INT} . Thus, the synaptic currents resulting from the afferent input to a region is a negligible compared to those generated by the activity in the local recurrent network. Thus, LFP-MP latency can be a simple method to determine the contribution of external inputs vs recurrent inputs in any brain region. These points are included in the revised manuscript's discussion section and the new Table 1.

A visual examination shows that similar effects exist in other publications too, though they did not address them:

- Saleem AB, Chadderton P, Apergis-Schoute J, Harris KD, Schultz SR. *Methods for predicting cortical UP and DOWN states from the phase of deep layer local field potentials. J Comput Neurosci. 2010 Aug;29(1-2):49-62. doi: 10.1007/s10827-010-0228-5. Epub 2010 Mar 12. PMID: 20225075; PMCID: PMC3094772.*

- Shows that LFP transitions to UP/DOWN states follows transitions detected in the membrane potential. They are not specific about the delay timing, but judging by their figures, LFP follows membrane potential by ~100 ms.
- Volgushev M, Chauvette S, Timofeev I. Long-range correlation of the membrane potential in neocortical neurons during slow oscillation. *Prog Brain Res.* 2011;193:181-99. doi: 10.1016/B978-0-444-53839-0.00012-0. PMID: 21854963; PMCID: PMC3397925.
 - Pairs of intracellularly recorded neurons up to ~1 cm apart exhibit delays of just a few ms. Anterior neurons preceded posterior neurons, consistent with hypothesis of travelling wave. Used this to calculate activity propagation at speed of 1-1.5 m/s.

We factor in this 'negative delay' to compute the net delay between the membrane potential of parietal neurons and the membrane potentials of neurons in all the other brain regions. All these latencies are positive and follow a systematic pattern expected by anatomy (parietal cortex being the input and other regions being the outputs). Thus, there is no conceptual problem or cause for concern. Furthermore, the latency is significantly different for down-to-up vs up-to-down transitions (Supp. Fig. 10)

2) Related to the previous point: what are the delays between the LFP and the neurons in frontal cortex? Do the authors also observe negative delays here (i.e., Vm fluctuations preceding the LFP)?

We have updated our figures (Main Fig. 4 and Sup. Fig 10, 11) to include frontal and prefrontal neuronal delays. When the LFP-MP latency within the parietal cortex is factored out, the resulting numbers are latencies between the Vm of different brain regions. The delays are positive between the Vm of parietal neurons and prefrontal neurons, i.e. parietal neuronal UDS comes first and then prefrontal after 40 ms. We have not measured the delay between LFP-MP within the frontal cortex. However, as discussed above, similar LFP-MP negative latency has been reported by others in several other brain areas too.

3) The model the authors propose is based on the Wilson-Cowan equations. In a recent paper [*], it was shown that the WC model does not accurately predict neuronal population responses to input pulses to excitatory and inhibitory populations, thus, the WC equations don't model cortex accurately. Do the authors think this could pose a problem for the application of the model in the present case?

[*] Lin, Okun, Carandini, & Harris (2020) Equations governing dynamics of excitation and inhibition in the mouse corticothalamic network. bioRxiv. doi: <https://doi.org/10.1101/2020.06.03.132688>

Lin et. al. examine the responses of a cortical network to short 1-4ms long laser pulses, which then trigger a barrage of activity lasting about 20 ms within the ontogenetically-altered excitatory population. They then observe that the ensuing activity in the broader cortex cannot be explained by the simple mean field equations of Wilson-Cowan or any similar mean-field model, and thus justify the inclusion a thalamic network within their model. Given how short the pulse duration is and the abrupt and transient triggered activity, it makes sense that a mean field model of with time scales as large as 300ms (adaptation) does not capture the full dynamics makes sense. Additionally, this optogenetic stimulation involves simultaneous activation of many synapses by an external force, which can create paradoxical results, e.g. high-conductance states (Vm reaching closer to the reversal potential of excitation in some neurons, causing nonlinear and even nonmonotonic synaptic activation, see Petersen et al. PNAS 2004.

Such external activation is neither created in our model and nor feasible to explore within the scope of our current model.

Our study, on the other hand, investigates sustained activity and inactivity, lasting 500-1000 ms. At these time scales. Finally, the proof is in the pudding --With just 2 networks and 2 free parameters we are able to explain a vast amount of experimental observations, unlike the findings of Harris et al.. Thalamic and other networks would of course contribute to the full network dynamics. But, the inherent simplicity of our model is actually one of its major strengths, as it can still reproduce quantitatively all the predictions we have tested.

4) The fraction of SPA/SPI seems to be largely identical for neurons in parietal and the neurons in frontal cortex and, as a result, the fitted model parameters are largely similar (based on the results obtained from the alternative fitting procedure shown in Supplementary Fig. 7). Considering that many differences have been reported between microcircuits in parietal and frontal cortical regions with respect to the level of recurrent excitation, the level of inhibition, as well as anatomical connectivity (e.g. Fulcher et al., 2019, PNAS; Wang, 2020, Nat Rev Neurosci), I am a bit surprised that the fitted model parameters (and the resulting dynamics) are so similar. Is this something to be expected or does it reflect the inability of the model to generalize beyond cortical-hippocampal interactions?

Fulcher et. al. infer large-scale gradients in cytoarchitecture, gene expression, cell densities, etc. using the MRI-derived T1w:T2w metric, commonly interpreted as a marker of intracortical myelin content. This ratio is known to be based on signal intensities that are inherently non-quantitative and affected by extrinsic factors.

On the other hand, anatomical studies like EPFL's Blue Brain Cell Atlas initiative (Reference 79) have produced precise measures of excitatory and inhibitory cell densities in various cortical regions. These studies can be more readily used to test our inferences on cortical statistics. Our model suggests that the internal excitation W_{INT} within PAR, FRO, PRE, and LECIII should be similar, and should collectively be significantly lower than in MECIII. This is indeed the case in the connectomics data, as shown in the revised Sup. Table 2, where the calculated value for excitation-inhibition cell density ratio obtained from the Blue-Brain and Allen-Brain connectome data is similar across neocortical (PAR and FRO) and LECIII populations. Remarkably the MECIII E:I ratio, for both dorsal and ventral regions, is much larger than the E:I ratio in LECIII, PAR, and FRO, also consistent with the model prediction. The study did not have any data from prefrontal areas.

All said, we share the referees sentiment that these results are surprising since there are many other differences across prefrontal and parietal, as well as between different neurons within these regions. Yet, the model is able to explain the results remarkably well. Perhaps this is because during UDS oscillations the system lives in a very low dimensional space where these difference do not make a difference, but these will become important during other behavioral states such as working memory. Or because the UDS in vivo involve a large number of other networks too, not just the two regions we investigated, and the collective dynamics reduces to just a two network model. We have clarified these points in the revised discussion section (See Discussion section, 3rd to last paragraph)

5) Somewhat related to the previous point, the model predicts that neurons with greater net excitatory input ($W_{ext} + W_{int}$) show higher firing rates, as confirmed by the recordings (with MECIII neurons showing higher firing rates than LECIII neurons). How do the firing rates look for the neurons in parietal and frontal cortex and how do they relate to the firing rates predicted by the model (based on the

estimated net excitatory input)? As the authors state in their discussion that the “(...) model is sufficiently general and could equally apply to other networks, e.g. parietal-prefrontal network (...)”, it would be important to see if the model makes accurate predictions for those areas. If not, this (and possible reasons for it) should be discussed.

These results are included in Supp. Fig 9, which has been updated from its previous iteration to include data from prefrontal and frontal neurons. Consistent with the predictions of our theory, prefrontal and frontal neurons also have lower firing rates compared to MECIII neurons. Thus, the firing rates of frontal neurons are consistent with the model predictions in all brain regions, even though we used only meanfield neurons without any explicit spiking.

Furthermore, analysis of the entire experimental population suggests a non-linear relationship between total excitation and firing rate, in line with several modeling and experimental studies.

6) On page 6, last paragraph, the authors write: “These latencies were more correlated with the predicted W_{int} and W_{ext} values than with simply the levels of SPA or SPI (Sup. Fig. 8), further supporting the model”. Looking at the figures, the differences in correlation values are not very pronounced. It seems to be that a statement like this requires a statistical comparison of the obtained correlation values.

We ran the analysis again using our updated simulation, and found that W_{EXT} was more anticorrelated (Sup. Fig 11Ai, $r=-0.66$, $p<10^{-16}$) with Down-Up delays than was the total SPI level (Sup. Fig. 11Aiii, $r=0.57$, $p<10^{-11}$). On the other hand, as predicted by the model there was no statistically significant difference between the correlations of Up-Down delays to W_{INT} levels vs. SPA levels (Sup. Fig 11Bi, 11Biii).

While this minimal model has many limitations, we are unaware of any other study that used a 2-free parameter mean field theory to obtain a quantitative match for fourteen different model predictions and experimental measurements in vivo.

Others

- Supplementary Figure 1: There seems to be a mistake in the figure caption related to panel A, where it says that the adaptation time constant is 500 s.

This has been fixed.

REVIEWERS' COMMENTS

Reviewer #1 (Remarks to the Author):

The authors have addressed my comments and I am satisfied with their responses.

Reviewer #2 (Remarks to the Author):

I would like to thank the authors for their thorough revision - all my concerns have been addressed. This is a strong paper that surely will find its audience.